# A universal language for finding mass spectrometry data patterns

Despite being information rich, the vast majority of untargeted mass spectrometry data are underutilized; most analytes are not used for downstream interpretation or reanalysis after publication. The inability to dive into these rich raw mass spectrometry datasets is due to the limited flexibility and scalability of existing software tools. Here we introduce a new language, the Mass Spectrometry Query Language (MassQL), and an accompanying software ecosystem that addresses these issues by enabling the community to directly query mass spectrometry data with an expressive set of user-defined mass spectrometry patterns. Illustrated by real-world examples, MassQL provides a data-driven definition of chemical diversity by enabling the reanalysis of all public untargeted metabolomics data, empowering scientists across many disciplines to make new discoveries. MassQL has been widely implemented in multiple open-source and commercial mass spectrometry analysis tools, which enhances the ability, interoperability and reproducibility of mining of mass spectrometry data for the research community.

Innovation in mass spectrometry (MS) has enabled tremendous progress in life sciences, and advances in MS instrumentation have led to the widespread adoption of omics disciplines (for example, metabolomics, lipidomics and proteomics). Despite the broad application of MS to characterize proteins, peptides, polymers, small molecules and nucleic acids across research disciplines, the ability for scientists to flexibly search for known chemical classes within and across MS datasets remains a challenge. The interrogation of MS data for the presence of specific chemicals or classes of molecules utilizes patterns in MS peaks representing intact analytes[1] isotopic signatures or characteristic mass differences (MS1), associated fragmentation patterns in tandem MS data (MS/MS), chromatographic retention time, collisional cross-section or combinations thereof. This search for specific patterns is usually performed through a slow and error-prone manual inspection of the data. Alternatively, specialized software tools have been developed for this purpose but are often limited to search for a specific compound[1] or a limited set of class-specific MS patterns[2]. Although bespoke one-off scripts provide the necessary flexibility to search for specific MS data patterns[3], most noncomputational researchers and laboratories lack the computational skills to develop or customize them[4,5]. This skill gap limits biologists and chemists from effectively searching across MS datasets, potentially leaving many biologically important molecules hidden and

undiscovered in the data. To address this gap, here we introduce the Mass Spectrometry Query Language (MassQL), an open-source language for flexible and mass spectrometer manufacturer-independent searching. MassQL aims to enable noncomputational researchers to easily search their MS (across MS1 and MS/MS) data for patterns of interest without the need for programming skills or a dedicated computational collaborator. In this Article, we describe the MassQL language, showcase its accompanying computational ecosystem to increase accessibility and highlight two application examples that demonstrate how MassQL can be used on entire public repositories (such as Global Natural Products Social Molecular Networking (GNPS)/MassIVE[6], Metabolomics Workbench[7] and MetaboLights[8]).

## Results

The versatility of MS to capture unique characteristics of chemical structures, such as isotopic patterns (for example, bromination), diagnostic fragmentation (for example, production of sulfur trioxide) and neutral losses (for example, loss of sugar moieties), makes it a powerful analytical tool but also presents challenges to effectively interpret and utilize the richness of the data. Specifically, the ability to simultaneously utilize some or all of these different dimensions of MS data in an integrated fashion is currently out of reach. To complement the versatility

✉e-mail: mingxun.wang@cs.ucr.edu

of MS data, the MassQL language implements a succinct and expressive grammar to search for chemically and biologically relevant molecules in the MS data by leveraging these patterns (Fig. 1a,b). The MassQL language enables searching for patterns in MS1 data (for example, isotopic patterns and adduct mass shift) and MS/MS fragmentation spectra (for example, presence/absence of fragments and neutral losses), as well as applying chromatographic and ion mobility constraints. In addition, MassQL provides language support for user-defined tolerances, such as ion intensity and mass accuracy (Fig. 1b). Moreover, each of these query elements can be combined with Boolean operators (for example, AND, OR and NOT) to form more complex queries. These properties and patterns are common to nearly all MS data types, thus making MassQL agnostic to the instrument vendor, mass detector (for example, Orbitrap and quadrupole time-of-flight), ionization source (for example, electrospray ionization and matrix assisted laser desorption/ionization) and separation method (for example, liquid chromatography, gas chromatography and ion mobility). Together, the MassQL language provides users the flexibility and expressiveness to query simple and complex MS patterns regardless of their computational expertise, thus lowering the barriers of entry to MS data interrogation. Finally, as a language, new MassQL terms can be defined, which enables grammar and syntax evolution to maintain compatibility of queries to advancing MS technologies.

The MassQL computational ecosystem is made of several key components that ensure its extensibility and usability for the community. These components are a formal definition of the MassQL grammar, a MassQL language parser, a reference implementation of the MassQL query engine, interactive web interfaces to enhance accessibility for noncomputational users and a NextFlow computational workflow for parallelized querying of very large datasets (Methods). Together, this enables MassQL searches on a single MS data file, within and across whole MS datasets, up to entire data repositories, including GNPS/MassIVE[6], Metabolomics Workbench[7] and MetaboLights[8].

The MassQL formal grammar builds upon common MS terminology (Methods), which makes MassQL queries easy to write and alter for scientists with basic familiarity with MS. To further help new users, we have created extensive documentation (https://mwang87.github.io/MassQueryLanguage_Documentation/), instructional videos (https://www.youtube.com/playlist?list=PLkDps_-pcYZ5D3rhas208dsMg66lCGmcs) and an interactive MassQL sandbox (https://massql.gnps2.org/). The sandbox enables users to interactively develop and test MassQL queries on demonstration data to check for queries' correctness before applying to their own data. Moreover, the sandbox automatically translates each query into English, Portuguese, Spanish, German, French, Mandarin Chinese, Japanese, Korean and Russian, which helps the interpretability of queries in manuscripts and grants. In addition, we have developed and deployed a large-language-model-powered conversational assistant (https://massql-analysis.gnps2.org/MassQL_Chatbot) to support new users with real-time MassQL query writing and troubleshooting. Finally, as a community effort, we created a wiki-like compendium of 35 applications of MassQL (https://massql.gnps2.org/compendium/; Fig. 1d) that serves as a reference and inspiration for new users. Queries in the compendium can be (re)used as they are, or as a starting point to develop queries tailored to the chemicals or compound classes of interest. The compendium is regularly updated with new examples of MassQL queries successfully used by community members in published works and will function as an 'app store' for a centralized MassQL query deposition and sharing.

While MassQL was originally implemented within the GNPS environment[6,9], this was limiting for the accessibility to a broader audience. To further enhance usability, we have engaged with the wider metabolomics software community and, through these efforts, the MassQL language has been adopted and natively supported in a variety of MS data analysis software and infrastructure (Fig. 1c), both open-source (MZmine[10], pyOpenMS[11], MS-DIAL[12], UniDec[13] and Metabolomics Workbench[7]) and commercial (Bruker's MetaboScape). It must be noted that these software tools use the same MassQL language grammar but implement their own MassQL query engine backend. This provides the possibility to develop optimized query engines for improved query performance, while maintaining query semantic consistency across tools. Finally, to facilitate the integration of MassQL into other platforms and pipelines, MassQL is available as Python and R (ref. 14) libraries and as a web application programming interface (API).

## Discovery of siderophores at repository scale

Iron-binding small molecules play essential roles across biology, including microbial or mammalian siderophores that facilitate iron homeostasis[15,16]. We recently developed a native metabolomics method that includes infusion of iron to identify iron-binding compounds from complex samples based on the identification of retention time and peak shape correlations along with an $Fe^{3+}$-characteristic $m/z$ delta of 52.91 (ref. 17). As a complementary strategy to our native metabolomics method, which requires hardware modification in instrument setup and data acquisition, we mined all existing public metabolomics data, encompassing over 230 million analytes, to discover putative iron-binding compounds using MassQL. Although iron is often stripped from iron-binding compounds in liquid chromatography, we hypothesized that some iron-binding compounds will remain bound to iron at detectable levels in the MS data.

To develop the MassQL query, we used an MS dataset collected from *Eutypa lata* supernatant extracts that were treated with a post-liquid chromatography iron addition, as described by Aron et al.[17] (Supplementary Notes 3.3.1 and 3.3.2). The refined MassQL query searches MS1 spectra for the characteristic isotopic pattern of iron along with a distinctive iron-binding mass shift (Fig. 2a). Specifically, the MassQL query searches for MS1 precursor ions with an $m/z$ of $x$, $m/z$ of $x - 1.993$ at 0.063% intensity of $x$ (the stable isotope ratio of $^{54}Fe$ to $^{56}Fe$), and $m/z$ of $x + 1.0034$ (the $^{13}C$ peak), in addition to a proton-bound adduct (apo) peak at $m/z$ of $x - 52.91$. Combined MassQL queries for apo and bound MS2 spectra identified seven out of the eight putative siderophores identified using ion-identity molecular networking (IIMN)[18] in the published analysis of the post-liquid chromatography iron addition of *E. lata* extracts. The unique compound that was found using IIMN but not by the MassQL query was missed because the $^{54}Fe$ peak intensity fell outside of the expected intensity tolerance of 25%, which is probably due to the low intensity of this peak. We used strict $m/z$ (10 ppm) and expected intensity percentage (25%) tolerances to minimize false-positive retrieval. Using MassQL, an additional four molecules were found that were not found using IIMN (Supplementary Fig. 3.3.2-1); these molecules are probably iron-binding, as manual inspection revealed that they exhibit the expected iron-bound isotopic pattern. The published IIMN analysis may have missed these molecules owing to requirements for peak shape and retention time correlations or owing to low-intensity peaks falling below feature finding thresholds.

After validation of the siderophore query on the *E. lata* dataset, we extended the search for iron-binding compounds to all public high-resolution Thermo Fisher Q Exactive data available in the GNPS/MassIVE repository[6] (Supplementary Note 3.3.3). In searching over 230 million MS/MS spectra in 97,109 public data files, we retrieved 26,944 MS/MS spectra associated with the iron-characteristic isotope pattern in their MS1 data. We used MS-Cluster[19] on the retrieved MS/MS spectra to collapse redundant observations of the candidate iron-binding molecules. This resulted in 7,504 consensus MS/MS spectra. Using these consensus spectra, we created a molecular network in GNPS. We could putatively identify 441 (5%) of the consensus spectra by spectral library search against the public MS/MS libraries in GNPS[6]. The putatively identified known compounds were further filtered to remove duplicates and adducts of ethylenediaminetetraacetic acid (a common anticoagulant). After filtering, 52% of annotated spectra are known iron binders, while an additional 25% are lipids, bile acids,

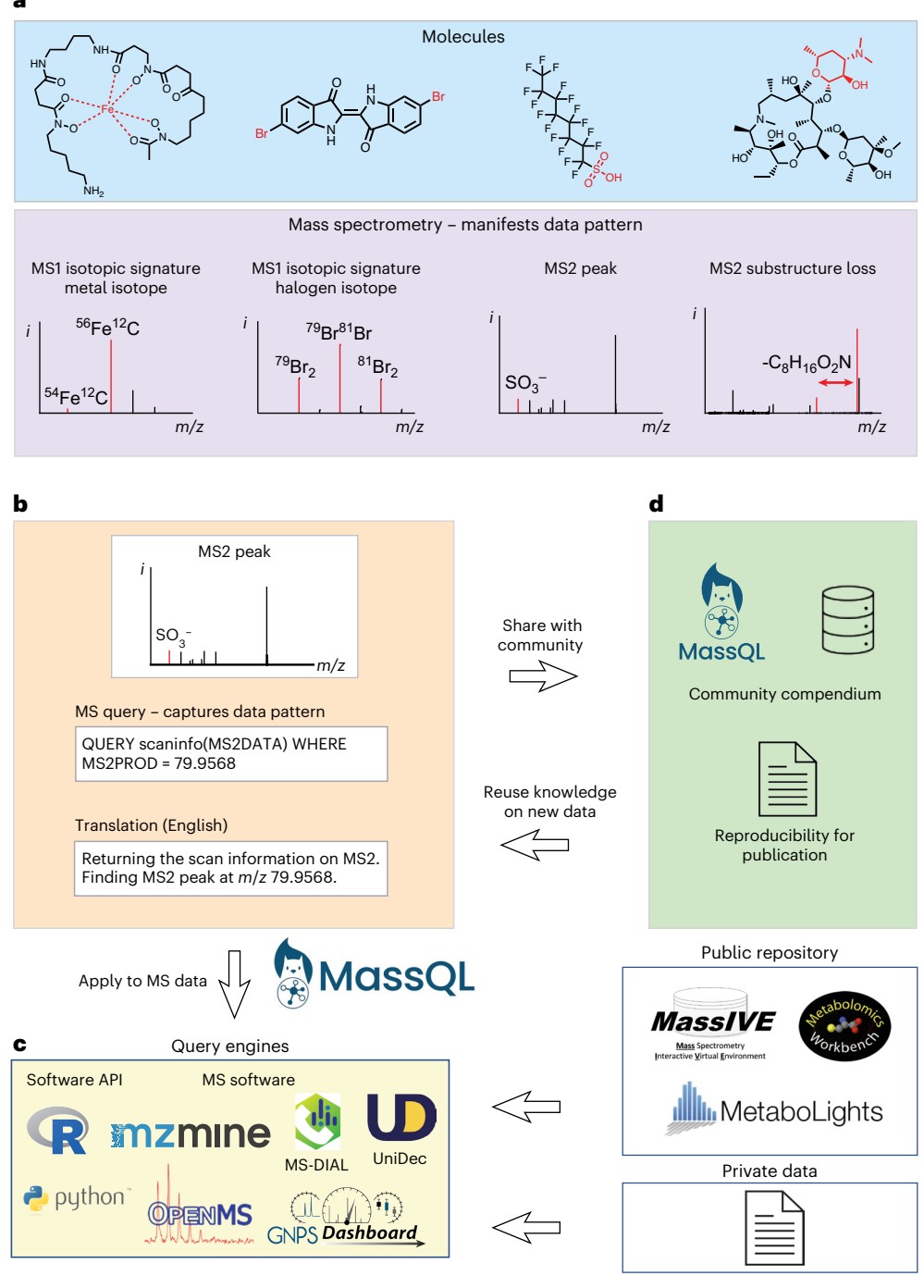

**Fig. 1 | Schematic representation of the MassQL ecosystem. a**, Examples of molecules that produce distinctive data patterns when measured by MS as mass/charge ($m/z$) and intensity ($i$) peaks. **b**, MassQL query representing MS/MS fragmentation patterns that encapsulates a characteristic mass loss. The query can be translated to nine languages for enhanced accessibility. **c**, MassQL is a universal tool to query MS data. MassQL enables data searching in a single file to entire MS repositories. MassQL has also been incorporated into a wide range of MS software. **d**, MassQL queries are shared and reused via the Community Compendium, which increases reproducibility and knowledge dissemination.

polyphenols and peptide/amino acids—classes that have been shown to bind iron and other divalent metals[20–23]. Notably, because such a large fraction (>95%) of the analytes in the molecular network could not be annotated to known molecules (Fig. 2d), this molecular network is probably a rich resource for the discovery of new siderophores.

### OPEs in the environments

Organophosphate esters (OPEs) are widely used flame retardants and plasticizers and are ubiquitously detected in environmental samples[24–26]. Recent studies in environmental chemistry have reported several novel OPEs by searching for a conserved, unique fragmentation pattern (O = P(OR)$_3$) in MS/MS data[27–29]. The characteristic fragment, frequently leveraged in literature, is the phosphate product ion (H$_4$O$_4$P$^+$, $m/z$ 98.9842). The traditional low-throughput methodologies, which includes manual analysis and targeted lists of known OPEs, limits the ability to discover novel potentially harmful chemicals. Here, we demonstrate how MassQL can enable high-throughput screening and discovery of novel OPE without a predefined suspect list.

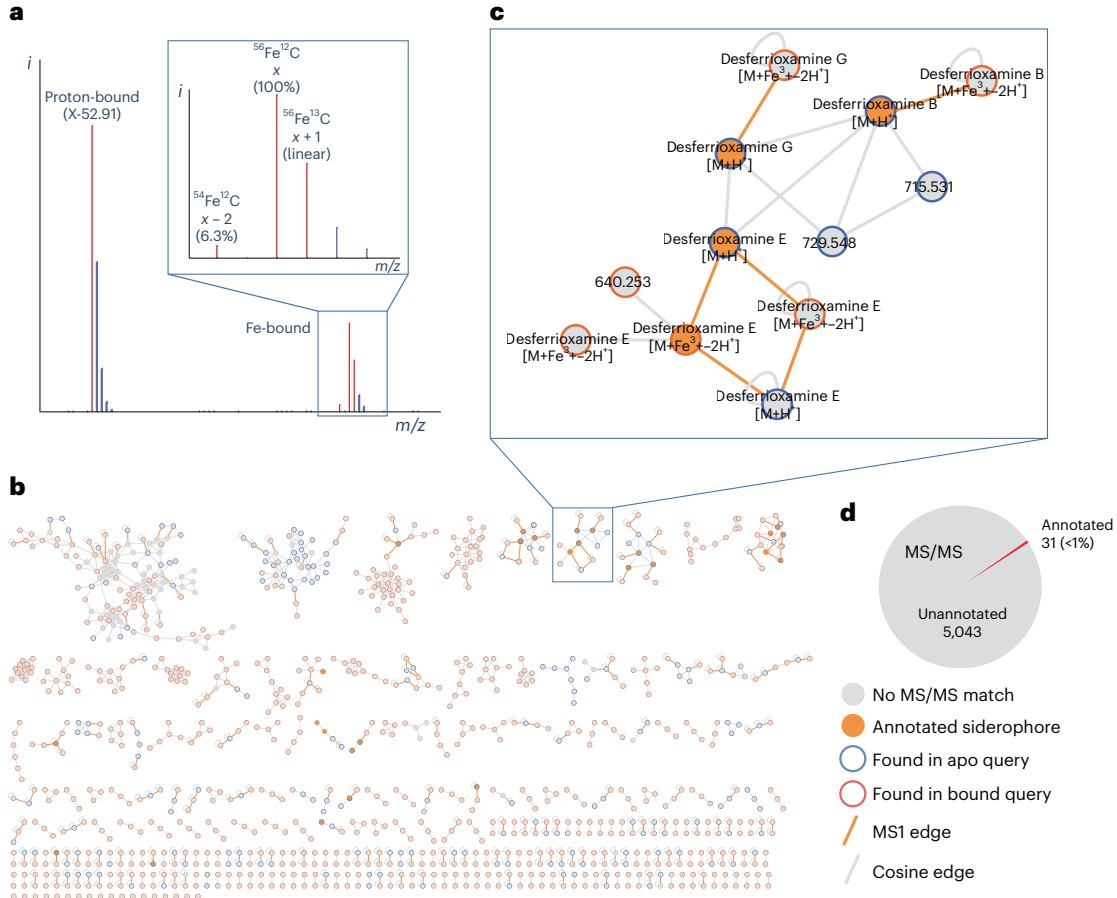

**Fig. 2 | Insights from siderophores at a repository scale. a**, The MassQL query for iron-binding searches for the characteristic $^{54}Fe^{12}C$ peak with 6.3% abundance relative to the $^{56}Fe^{12}C$ peak. Peaks queried by MassQL are colored in red. **b**, The molecular network of MassQL spectra hits after clustering by MSCluster (gray, no MS/MS annotation by GNPS MS/MS library; orange, MS/MS annotation by GNPS MS/MS libraries). Singletons (no neighbors in the network) have been excluded from this molecular network. **c**, A spectral family of the molecular network containing desferrioxamines, including proton-bound and iron-bound desferrioxamines E, G and B, in addition to structurally related analogs. **d**, Less than <1% of clustered MS/MS are annotated as siderophores by GNPS libraries when masses associated with ethylenediaminetetraacetic acid (an anticoagulant added to MS samples) are removed from the network.

Utilizing the characteristic phosphate product ion, we formulated a MassQL query to search for an MS/MS peak at $m/z$ 98.9847 with 50 ppm mass error tolerance and a peak intensity >50% of the base peak (Supplementary Notes 3.2.3 and 3.2.4). Similarly to siderophores example, a MassQL query was developed on a test marine water dataset to verify the utility of the MassQL for identifying putative OPEs in complex samples (Supplementary Note 3.2.3). In this test dataset, where three OPEs were previously identified by manual analysis[30], MassQL returned 589 MS/MS spectra belonging to ~60 unique molecular features. MS/MS library search against the GNPS spectral library of the MassQL retrieved MS/MS spectra putatively annotated four OPE molecules, including all three previously identified OPEs, and one new putatively identified OPE. We additionally putatively identified two non-OPE molecules. The first molecule contained a phosphate group that resulted in the characteristic phosphate fragment in the MassQL query. The second putative non-OPE molecule did not contain a phosphate group, but further investigation revealed a potentially false-positive library match due to large mass errors in the library MS/MS peaks. These results suggested that the MassQL query was useful in retrieving OPEs but may also capture a broader range of phosphate-containing compounds. For this reason, when scaling up to repository searches (see below), we leverage the ability of molecular networking to group together similar structures and segregate OPEs from phosphate-containing compounds more generally.

To identify OPEs in public data, we scaled the MassQL query to all Q Exactive data in the GNPS/MassIVE[6] data repository (which included >230 million MS/MS spectra). The MassQL query found 338,439 MS/MS matching the query criteria. Only 15% (51,310) of the MS/MS found by MassQL could be explained (precursor $m/z$ match with 20 ppm mass error) by known OPEs based on a comprehensive OPEs list ($n$ = 95) compiled by Ye et al.[29]. We extracted all MS/MS spectra and created consensus MS/MS spectra using Falcon-MS[30], resulting in 2,777 consensus spectra. We used these consensus spectra to create a molecular network. Combining the library annotation results, we propose one additional OPE that was not included in the GNPS library and the comprehensive OPE list by Ye et al.[29] (Fig. 3). It is important to reemphasize that, in the search for OPEs, the MassQL query was not designed to specifically look for OPEs but phosphate-containing molecules more generally. The molecular networking strategy complemented the MassQL results to organize OPE molecules in their own families. This combination of MassQL and molecular networking was critical because MassQL greatly reduced the data size (from 230 million to ~338,000 MS/MS, making molecular networking computationally tractable), while molecular networking helped to focus attention on families of specifically organophosphate molecules.

### False discovery rate estimation and query validation strategies
Overall, the key challenge when using MassQL is to define queries that are sensitive toward the target compound(s) (that is, effectively retrieve

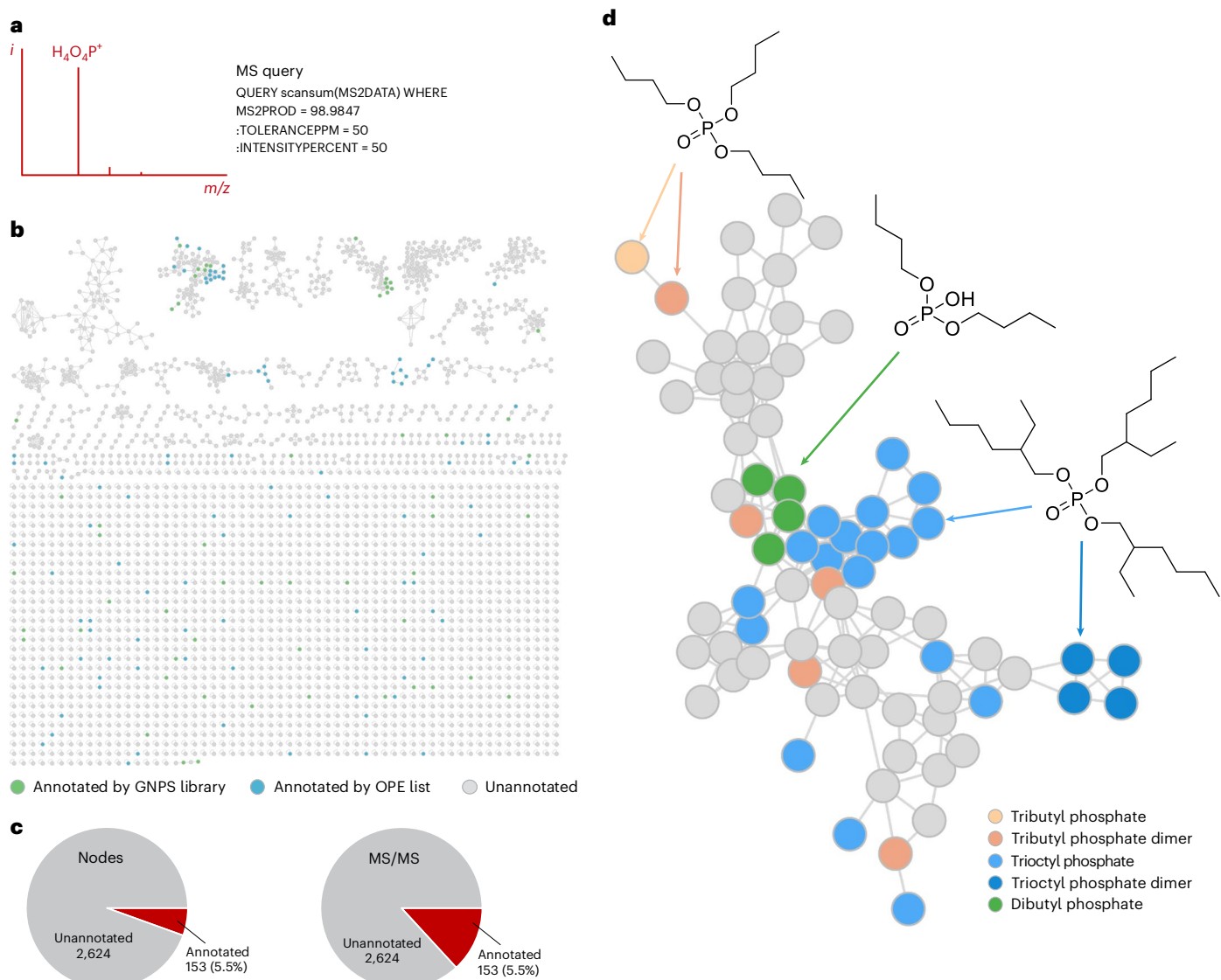

**Fig. 3 | Discovering OPEs at a repository scale. a**, The general structure of OPEs and the MassQL query for the characteristic phosphate fragment. **b**, The molecular network of MassQL MS/MS in a repository scale query after clustering by Falcon[30] (green, annotation by GNPS library; blue, annotation by an OPEs list curated by Ye et al.[29]; gray, unannotated). **c**, Summary of MS matching results (precursor *m/z* match with 20 ppm mass error) by the GNPS MS/MS library and the OPEs list by Ye et al.[29]. These putative identifications were based on precursor only (level 3 annotations). **d**, A molecular family shown containing alkyl-OPEs. OPEs reported by Ye et al. or in the MS/MS database search are indicated: tributyl phosphate and dimers (light orange and dark orange), and trioctyl phosphate and dimers (light blue and dark blue). Dibutyl phosphate (green) was not reported by Ye et al. or in the MS/MS database search. The structures displayed are illustrative of one possible isomer of the alkyl chain; the specific structure is beyond the scope of this report.

the desired spectra) but do not retrieve too many false-positive hits. Due to the flexibility and broad applicability of MassQL, a universal method for false discovery rate (FDR) estimation is difficult to establish. Rather, tailored strategies for query validation can be designed case by case by the user, depending on the research context.

As demonstrated in a recent publication that used MassQL to mine liquid chromatography–MS/MS raw data in the public domain and discover new, unreported bile acids[31], one possible strategy to estimate the FDR of MassQL queries over MS/MS spectra is to putatively identify the MS/MS spectra retrieved by MassQL using reference MS/MS spectral libraries. In their study, Mohanty et al. first used the GNPS spectral libraries (which contained 4,533 reference spectra of bile acids) to design and refine MassQL queries for bile acid spectra and estimate the queries' selectivity. Such selectivity was measured by counting the number of retrieved bile acids and the number of retrieved non-bile acids (false positives). Thereafter, when using the refined MassQL

queries to search repository data, more than 594,000 putative bile acids MS/MS spectra were retrieved. Among these, 270,437 MS/MS spectra were putatively identified by MS/MS library search, with 726 MS/MS matching to non-bile acids (0.27%). It is important to highlight that this FDR estimation approach is limited to the compounds that are deposited in the reference MS/MS libraries.

It should be noted that the same validation approach cannot be universally applied, for example, to the repository-scale discovery of siderophores described in the present manuscript ('Discovery of siderophores at repository scale'). Siderophore molecules do not belong to a single compound class and can exhibit very diverse chemical structures. Therefore, queries for diagnostic MS/MS fragment ions cannot be used, and the search was instead performed on the MS1 level ('Discovery of siderophores at repository scale' section). The adopted strategy was to first develop and refine the query on a reference dataset of *E. lata* extracts known to contain iron-binding molecules[17]. After

satisfactory results were obtained on the reference dataset (Supplementary Note 3.3.2), the query was performed on a repository scale. In this specific example, we used a strategy based on a 'decoy' query to get a sense for potential false discoveries (Supplementary Note 2), estimated at 21.7% for this repository scale query. Although a relatively large number of false-positive hits may be expected when performing searches at a repository scale, these false positives can be mitigated by utilizing additional computational tools in a downstream analysis (for example, molecular networking and spectral library search) to validate the query results. In these cases, MassQL acts more as a prefilter to reduce the data to a more tractable size (from hundreds of millions of spectra to a few thousands) and 'enrich' them with putative leads for further investigation and confirmatory experiments ('Discovery of siderophores at repository scale' and 'OPEs in the environments' sections). Overall, we encourage users to critically inspect query results and develop tailored validation strategies that are fit for their research purposes.

## Discussion

Here, we introduced MassQL, a platform- and manufacturer-independent query language to search for MS data patterns within and across MS datasets. The main goal of MassQL is to provide noncomputational scientists with the flexibility to encode complex MS patterns into concise and expressive queries, reducing the need to write bespoke programming scripts. Together with the accompanying software infrastructure created through community efforts, the MassQL ecosystem lowers the barrier of entry to MS data interrogation for chemists and biologists lacking programming skills. This goal is being realized by the adoption of MassQL by the community. Of note, MassQL has already been used by researchers for MS data mining[32], microbiome research[31], exposomic and biomonitoring[33], infectious disease research[34] and natural product discovery[35–42].

While traditional MS/MS similarity/search tools are a powerful technique for matching related or similar compounds within libraries or repositories (for example, spectral library search[43] and MASST[44]), MassQL provides a complementary set of capabilities with enhanced flexibility and precision. Specifically, traditional MS/MS search tools rely on a full MS/MS similarity measure to retrieve a match, whereas MassQL queries are a set of user-defined constraints to retrieve a spectrum. This provides the flexibility to search for more specific and complex patterns (for example, combine MS1 and MS/MS patterns, retention and drift time constraints; Supplementary Notes 3.3 and 3.4) and empowers scientists to leverage their domain knowledge of the chemical or compound class under investigation. A specific example where MassQL can complement MS/MS similarity is when small structural modifications can result in large changes in the overall fragmentation patterns. This situation can cause relevant analog molecules to evade discovery by MS/MS similarity-based search. MassQL has been shown to complement MS/MS comparison strategies by enabling the searching for conserved key fragments or neutral losses in the MS/MS spectrum without requiring a full MS/MS similarity match (for example, ref. [42]).

While MassQL-based querying of small and large datasets can be an effective way to prioritize data, the utility of this querying paradigm can be enhanced when paired with complementary analysis tools (before or after MassQL). First, in this Article, we showcased how discovery can be enhanced by combining upstream MassQL searches with downstream molecular networking and spectral library analysis (see 'Discovery of siderophores at repository scale' and 'OPEs in the environments' sections in the Results). The use of MassQL as a prefiltering tool was essential in making the analysis possible, both from a computational tractability and data interpretability/prioritization perspective. Second, MassQL has been shown in the literature as a downstream tool to enhance the analysis of molecular networking—specifically, to aid in the prioritization of relevant compounds and, as highlighted above, to

overcome shortcomings in MS/MS alignment[35,37–39,41,42]. We do acknowledge limitations with MassQL as available today—specifically, MassQL has limited capabilities to leverage more than a handful of MS spectra, for example, consecutive MS spectra arising from the elution of chromatographic peaks that can be grouped as a chromatographic feature.

We envision that the MassQL computational ecosystem will grow in adoption and capability. We recognize areas for improvement within MassQL, such as enhancing the query performance of MassQL and expanding the expressiveness of the language. We have designed and scaffolded the MassQL ecosystem to be extensible in both of these respects. We have defined a format context-free grammar for MassQL that is separate from any query engine that implements the MassQL semantics (Methods). Even though the reference implementation of the MassQL query engine is not explicitly optimized for speed (Supplementary Note 1), this architecture enables the wider community to develop new query algorithms to improve search speed, while using the same formal grammar. This paradigm has already been demonstrated in the popular software tools that implemented their own query engine: MS-DIAL[12], Mzmine[10] and Bruker's MetaboScape. Finally, MassQL derives its strength from a vibrant user open-source community, and we expect this community-guided evolution of the language to continue in the future. As both the language and software ecosystem evolve, MassQL will become more capable and versatile to meet the scientific community's growing needs in mining MS data.

## Online content

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

Tito Damiani ®[1,59], Alan K. Jarmusch ®[2,59], Allegra T. Aron[3,59], Daniel Petras[4,5,59], Vanessa V. Phelan[6,59], Haoqi Nina Zhao[7,59], Wout Bittremieux[7], Deepa D. Acharya[8], Mohammed M. A. Ahmed ®[9,10], Anelize Bauermeister[11], Matthew J. Bertin[12], Paul D. Boudreau[9], Ricardo M. Borges[13], Benjamin P. Bowen ®[14,15], Christopher J. Brown ®[16], Fernanda O. Chagas[13], Kenneth D. Clevenger[17], Mario S. P. Correia[18], William J. Crandall ®[19], Max Crüsemann[20,21], Eoin Fahy ®[22], Oliver Fiehn ®[23], Neha Garg ®[24], William H. Gerwick[7,25], Jeffrey R. Gilbert[16], Daniel Globisch ®[18], Paulo Wender P. Gomes[26], Steffen Heuckeroth ®[27], C. Andrew James[28], Scott A. Jarmusch ®[29], Sarvar A. Kakhkhorov ®[30], Kyo Bin Kang ®[31], Nikolas Kessler[32], Roland D. Kersten[33], Hyunwoo Kim ®[34], Riley D. Kirk[35], Oliver Kohlbacher ®[36], Eftychia E. Kontou[37], Ken Liu[19], Itzel Lizama-Chamu[38], Gordon T. Luu[38], Tal Luzzatto Knaan[39], Helena Mannochio-Russo[7], Michael T. Marty ®[40], Yuki Matsuzawa[41], Andrew C. McAvoy[42], Laura-Isobel McCall[43], Osama G. Mohamed ®[44,45], Omri Nahor[39], Heiko Neuweger[32], Timo H. J. Niedermeyer ®[46], Kozo Nishida[41], Trent R. Northen ®[14,15], Kirsten E. Overdahl[2], Johannes Rainer ®[47], Raphael Reher ®[48], Elys Rodriguez[23], Timo T. Sachsenberg[49], Laura M. Sanchez ®[38], Robin Schmid ®[1,7], Cole Stevens[50], Shankar Subramaniam[22], Zhenyu Tian ®[51], Ashootosh Tripathi ®[33,45], Hiroshi Tsugawa ®[41,52,53], Justin J. J. van der Hooft ®[54,55], Andrea Vicini[47], Axel Walter[49], Tilmann Weber ®[37], Quanbo Xiong[56], Tao Xu[57], Tomáš Pluskal ®[1], Pieter C. Dorrestein ®[7] & Mingxun Wang ®[58] ✉

[1]Institute of Organic Chemistry and Biochemistry of the Czech Academy of Sciences, Prague, Czech Republic. [2]Metabolomics Core Facility, Immunity, Inflammation, and Disease Laboratory, Division of Intramural Research, National Institute of Environmental Health Sciences, National Institutes of Health, Research Triangle Park, NC, USA. [3]Department of Chemistry and Biochemistry, University of Denver, Denver, CO, USA. [4]Functional Metabolomics Lab, CMFI Cluster of Excellence, University of Tuebingen, Tuebingen, Germany. [5]Department of Biochemistry, University of California Riverside, Riverside, CA, USA. [6]Department of Pharmaceutical Sciences, Skaggs School of Pharmacy and Pharmaceutical Sciences, University of Colorado Anschutz Medical Campus, Aurora, CO, USA. [7]Collaborative Mass Spectrometry Innovation Center, Skaggs School of Pharmacy and Pharmaceutical Sciences, University of California San Diego, La Jolla, CA, USA. [8]Biologicals and Natural Products Discovery, Crop Protection R&D, Corteva Agrisciences, Indianapolis, IN, USA. [9]BioMolecular Sciences, School of Pharmacy, University of Mississippi, Oxford, MS, USA. [10]Pharmacognosy, Faculty of Pharmacy, Al-Azhar University, Nasr City, Egypt. [11]Department of Fundamental Chemistry, Institute of Chemistry, University of São Paulo, São Paulo, Brazil. [12]Department of Chemistry, Case Western Reserve University, Cleveland, OH, USA. [13]Walter Mors Institute of Research on Natural Products, Federal University of Rio de Janeiro, Rio de Janeiro, Brazil. [14]Environmental Genomics and Systems Biology Division, Lawrence Berkeley National Lab, Berkeley, CA, USA. [15]The Joint Genome Institute, Lawrence Berkeley National Lab, Berkeley, CA, USA. [16]Mass Spectrometry Center of Expertise, Regulatory and Stewardship, Corteva Agrisciences, Indianapolis, IN, USA. [17]Biologicals and Natural Products, Crop Protection R&D, Corteva Agrisciences, Indianapolis, IN, USA. [18]Department of Chemistry – BMC, Science for Life Laboratory, Uppsala University, Uppsala, Sweden. [19]Clinical Biomarkers Laboratory, School of Medicine, Emory University, Atlanta, GA, USA. [20]Institute of Pharmaceutical Biology, University of Bonn, Bonn, Germany. [21]Institute of Pharmaceutical Biology, Goethe University Frankfurt, Frankfurt, Germany. [22]Department of Bioengineering, University of California San Diego, La Jolla, CA, USA. [23]West Coast Metabolomics Center, University of California Davis, Davis, CA, USA. [24]School of Chemistry and Biochemistry, Center for Microbial Dynamics and Infection, Georgia Institute of Technology, Atlanta, GA, USA. [25]Scripps Institution of Oceanography and Skaggs School of Pharmacy and Pharmaceutical Sciences, University of California San Diego, La Jolla, CA, USA. [26]Faculty of Chemistry, Institute of Exact and Natural Science, Federal University of Para, Belem, Brazil. [27]Institute of Inorganic and Analytical Chemistry, University of Münster, Münster, Germany. [28]Center for Urban Waters, University of Washington, Tacoma, WA, USA. [29]Department of Biotechnology and Biomedicine, Technical University of Denmark, Kongens Lyngby, Denmark. [30]Laboratory of Physical and Chemical Methods of Research, Center for Advanced Technologies, Tashkent, Uzbekistan. [31]College of Pharmacy, Sookmyung Women's University, Seoul, Republic of Korea. [32]SW R&D Bioinformatics, Life Science Mass Spectrometry, Bruker Daltonics GmbH & Co. KG, Bremen, Germany. [33]Department of Medicinal Chemistry, College of Pharmacy, University of Michigan, Ann Arbor, MI, USA. [34]College of Pharmacy and Integrated Research Institute for Drug Development, Dongguk University-Seoul, Goyang, Republic of Korea. [35]College of Pharmacy, University of Rhode Island, Kingston, RI, USA. [36]Applied Bioinformatics, Department of Computer Science, University of Tuebingen, University of Tuebingen; Institute for Bioinformatics and Medical Informatics, University of Tuebingen; Institute for Translational Bioinformatics, University Hospital Tuebingen, Tübingen, Germany. [37]The Novo Nordisk Foundation Center for Biosustainability, Technical University of Denmark, Kongens Lyngby, Denmark. [38]Department of Chemistry and Biochemistry, UC Santa Cruz, Santa Cruz, CA, USA. [39]Department of Marine Biology, The Leon H. Charney School of Marine Sciences, University of Haifa, Haifa, Israel. [40]Department of Chemistry and Biochemistry, University of Arizona, Tucson, AZ, USA. [41]Department of Biotechnology and Life Science, Tokyo University of Agriculture and Technology, Koganei, Japan. [42]School of Chemistry and Biochemistry, Georgia Institute of Technology, Atlanta, GA, USA. [43]Department of Chemistry and Biochemistry, San Diego State University, San Diego, CA, USA. [44]Pharmacognosy Department, Faculty of Pharmacy, Cairo University, Cairo, Egypt. [45]Natural Products Discovery Core, Life Sciences Institute, University of Michigan, Ann Arbor, MI, USA. [46]Institute of Pharmacy, Freie Universität Berlin, Berlin, Germany. [47]Institute for Biomedicine, Eurac Research, Bolzano, Italy. [48]Department of Pharmacy, University of Marburg, Marburg, Germany. [49]Applied Bioinformatics, Department of Computer Science, University of Tuebingen, University of Tuebingen, Tübingen, Germany. [50]Department of BioMolecular Sciences, School of Pharmacy, University of Mississippi, Oxford, MS, USA. [51]Chemistry and Chemical Biology, Northeastern University, Boston, MA, USA. [52]RIKEN Center for Integrative Medical Sciences, Tsurumi-ku, Japan. [53]RIKEN Center for Sustainable Resource Science, Tsurumi-ku, Japan. [54]Bioinformatics Group, Wageningen University & Research, Wageningen, the Netherlands. [55]Department of Biochemistry, University of Johannesburg, Johannesburg, South Africa. [56]Crop Protection R&D, Corteva Agrisciences, Indianapolis, IN, USA. [57]Data Science and Bioinformatics, Corteva Agrisciences, Dublin, OH, USA. [58]Department of Computer Science, University of California Riverside, Riverside, CA, USA. [59]These authors contributed equally: Tito Damiani, Alan K. Jarmusch, Allegra T. Aron, Daniel Petras, Vanessa V. Phelan, Haoqi Nina Zhao. ✉e-mail: mingxun.wang@cs.ucr.edu

## Methods

### MassQL language description

**MassQL condition description.** The following table gives a short summary of the most common ways that we can set a condition to find the data we want. Each of the following conditions may have more than one way to qualify each condition, to modulate the tolerance, intensity and so on to help improve the specificity a user desires. Further details can be found in the official documentation at https://mwang87.github.io/MassQueryLanguage_Documentation/.

| Data type | MassQL syntax | Example |
|---|---|---|
| MS1 peak *m/z* | MS1MZ=<m/z value> | MS1MZ=163.1 |
| MS2 precursor *m/z* | MS2PREC=<m/z> | MS2PREC=488.1 |
| MS2 precursor charge | CHARGE=<Value> | CHARGE=2 |
| Fragmentation product ion *m/z* | MS2PROD=<m/z> | MS2PROD=163.1 |
| Ionization polarity | POLARITY=<Value> | POLARITY=Positive |
| Retention time (minimum) | RTMIN=<Value in Minutes> | RTMIN=5 |
| Retention time (maximum) | RTMAX=<Value in Minutes> | RTMAX=10 |
| Scan number (minimum) | SCANMIN=<Value> | SCANMIN=5 |
| Scan number (maximum | SCANMAX=<Value> | SCANMAX=5 |
| Ion mobility | MOBILITY=range (min=<min>, max=<max>) | MOBILITY=range (min=1, max=2) |

**MassQL qualifier description.** The following gives a short summary of the most common qualifiers for conditions that users can specify in MassQL. Further details can be found in the official documentation at https://mwang87.github.io/MassQueryLanguage_Documentation/.

| Qualifier type | Condition | Example |
|---|---|---|
| *m/z* tolerance | MS2PROD, MS1MZ, MS2PREC | MS2PROD=163.1:TOLERANCEMZ=0.1 |
| *m/z* ppm tolerance | MS2PROD, MS1MZ, MS2PREC | MS2PROD=163.1:TOLERANCE PPM=50 |
| Peak intensity minimum | MS2PROD, MS1MZ | MS2PROD=163.1:INTENSITYVALUE=1000 |
| Peak intensity minimum percent of base peak | MS2PROD, MS1MZ | MS2PROD=163.1:INTENSITYPERCENT=10 |
| Peak intensity minimum percent of TIC | MS2PROD, MS1MZ | MS2PROD=163.1:INTENSITYTICPERCENT=10 |
| Mass defect of peak | MS2PROD, MS1MZ, MS2PREC | MS2PROD=ANY:MASSDEFECT=mass defect (min=0.1, max=0.2) |
| Exclusion of condition | MS2PROD, MS1MZ, MS2PREC | MS2PROD=163.1:EXCLUDED |

### MassQL reference implementation

The reference implementation is a fully working version of the MassQL software ecosystem for the community to use. It also serves as a guide for future MassQL implementations that may optimize speed and/or introduce new functions in other systems. Specifically, the reference implementation includes the following pieces:

- MassQL formal grammar. The grammar is defined using the extended backus-naur form and builds upon common MS terminology for improved expressiveness and interpretability (see 'MassQL language description' section). During the development, community input and feedback shaped the vocabulary and capabilities of the language.

- MassQL parser. The parser transforms a query into an internal data structure that can be used by any programming language. The parsing is done by using the lark Python library (https://github.com/lark-parser/lark) and specific Python code to transform a MassQL query to a parse tree and into the internal data structure that organizes all query conditions and qualifiers.

- MassQL query engine. The MassQL reference query engine is written in Python and utilizes pyteomics[45] to read open MS data files from mzML, mzXML and MGF formats into data frames. Such MS spectra in data frame format can optionally be saved as Apache feather files to cache data for repeated querying. The query engine itself processes the query over these data frames using the Python pandas library to perform data filtering and manipulations. Output results are data frames that can be exported as a tabular format. Optionally, the retrieved MS spectra can be exported in JSON format, MGF and mzML[46].

The MassQL reference NextFlow[47] workflow is designed as an automated high-throughput tool for querying multiple files simultaneously on a computational cluster. This workflow utilizes the reference query engine and parallelizes the querying of multiple files across a multicore processor or a batch cluster, depending on the compute resources available. All results are then merged together, including extracted MS spectra in JSON format, MGF and mzML format, if desired.

### Reporting summary

Further information on research design is available in the Nature Portfolio Reporting Summary linked to this article.

## Data availability

All data used in this paper are deposited at MassIVE (massive.ucsd.edu). The relevant dataset accessions are provided together with the relevant description in the Supplementary Notes.

## Code availability

Reference Engine Implementation (Python), language formal grammar, GNPS Workflow, NextFlow Workflow and interactive web interface are available via GitHub at https://github.com/mwang87/MassQuery Language and Zenodo at https://doi.org/10.5281/zenodo.14419767 (ref. 48). MassQL is also available via Python API (https://pypi.org/project/massql/), R API (https://github.com/rformasssspectrometry/SpectraQL), mzmine (https://github.com/mzmine/mzmine), OpenMS (https://pyopenms.readthedocs.io/en/latest/massql.html), MS-DIAL 5 (http://prime.psc.riken.jp/compms/index.html) and UniDec (https://github.com/michaelmarty/UniDec). The official online documentation for MassQL is available at https://mwang87.github.io/MassQueryLanguage_Documentation/.

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

## Acknowledgements

We thank A. Leung for initial discussions and guidance on language design. This research was supported in part by the BBSRC-NSF award 2152526 (Dorrestein), the Intramural Research Program of National

Institute of Environmental Health Sciences of the NIH (ES103363-01, Jarmusch), the National Institute of General Medical Sciences of the NIH (R01 GM125943, Sanchez; R01 GM107550, Gerwick/Dorrestein; R35GM128690, Phelan), the National Institute of Allergy and Infectious Diseases of the NIH (R21AI156669, McCall; R15AI137996, Stevens), National Science Foundation (2128044, Sanchez; CHE-1845230, Marty), NSF CAREER Award (2047235 Garg), the Burroughs Wellcome Fund (1021280, McCall), Fundação de Amparo à Pesquisa do Estado de São Paulo (2018/24865-4), Fundação de Amparo à Pesquisa do Estado do Rio de Janeiro (E-26/201.260/2021, Borges; E-26/211.314/2019, Chagas), NIH (R35GM146934, Kersten), National Research Foundation of Korea (NRF-2020R1C1C1004046, RS-2022-NR068419, RS-2022-NR070845, and RS-2024-00436674, Kang), NIH U2C-DK119886 and NIH U24-DK141185 (Subramaniam), NIH R01 GM155383 (Fiehn), the German Research Foundation (EXC 2124, Petras), the Swedish Research Council (VR 2020-04707, Globisch), Fund for Financing Science and Innovation Support under the Ministry of Innovative Development of the Republic of Uzbekistan (Kakhkhorov), the National Research Foundation of Korea (NRF-2020R1C1C1004046, Kang; NRF-RS-2023-00211868, Kim), the German Research Foundation (DFG) TRR 261 (project 398967434, Walter), Biological Sciences Initiative at the University of Michigan (Tripathi), Heisenberg program (project 495740318, Crüsemann), the German Ministry for Education and Research de.NBI, BMBF FKZ031 A 534A and EPIC-XS, project number 823839, Federal Ministry of Education and Research in the frame of de.NBI/ELIXIR-DE (W-de.NBI-022), the Ministry of Science, Research and Arts Baden-Württemberg and EPIC-XS, project number 823839, funded by the Horizon 2020 program of the European Union (Kohlbacher, Sachsenberg), the Czech Science Foundation (GA CR, project number 21-11563M, Pluskal), the European Regional Development Fund (P JAC, project number CZ.02.01.01/00/22_010/0002733, Damiani), US Department of Energy Joint Genome Institute (https://ror.org/04xm1d337; a DOE Office of Science User Facility, is supported by the Office of Science of the US Department of Energy operated under contract number DE-AC02-05CH11231 (Northen and Bowen) and subcontract number 7601660, Wang), JSPS KAKENHI (21K18216, Tsugawa), the National Cancer Center Research and Development Fund (2020-A-9, Tsugawa), JST ERATO Grant (JPMJER2101, Tsugawa), AMED Japan Program for Infectious Diseases Research and Infrastructure (21wm0325036h0001, Tsugawa), JST NBDC (JPMJND2305, Tsugawa), the Novo Nordisk Foundation, Denmark (NNF20CC0035580, NNF16OC0021746, Weber), Betty and Gordon Moore Foundation (Aron), NIH (R35GM155026-01, Aron), DNRF137 (S. Jarmusch), NIH (5U24DK133658-02, Wang) and NIH (R35GM128690, Phelan) and the ALSAM Foundation (Phelan). M.W. was partially supported by the US Department of Energy Joint Genome Institute (https://ror.org/04xm1d337), a DOE Office of Science User Facility, and is supported by the Office of Science of the US Department of Energy under contract no. DE-AC02-05CH11231.

## Author contributions

M.W. conceived the project. M.W. and A.K.J. designed the language. M.W., R.S., J.R., A.V., H.T., M.T.M., T.T.S., N.K. and H.N. developed the software. M.W. and P.C.D. supervised the development of the project. D.P., A.T.A., A.K.J., A.B., M.T.M., W.B., N.G., V.V.P. and P.C.D. provided feedback. A.B., J.J.J.v.d.H., K.B.K., L.I.M., R.S., M.T.M., S.A.K., T.P., W.B., R.M.B., F.O.C. and Q.X. tested the software. A.B., J.J.J.v.d.H., K.B.K., K.L., L.I.M., W.C., T.D., T.P., M.S.P.C., D.G., A.C.M., N.G., M.C., M.J.B., R.M.B., F.O.C., O.G.M., A.T., E.E.K., T.W., M.M.A.A., P.D.B., S.H., Q.X., A.T.A., D.D.A., E.F., O.F., W.H.G., J.R.G., C.A.J., R.D.K., H.K., I.L.C., G.T.L., T.L.K., H.M.R., Y.M., O.N., T.H.J.N., K.N., T.R.N., K.E.O., R.R., E.R., L.M.S., C.S., S.S., Z.T., H.N.Z., and A.W. contributed a use case. M.W., A.T.A. and D.P. wrote the documentation. T.D., M.W., A.T.A., H.N.Z., A.K.J., V.V.P. and P.C.D. wrote the paper. All authors edited and approved of the paper.

## Competing interests

P.C.D. is an advisor to Cybele and a co-founder and scientific advisor to Ometa and Enveda with prior approval by UC San Diego. M.W. is a co-founder of Ometa Labs LLC. R.S., S.H. and T.P. are co-founders of mzio GmbH. T.R.N. is an advisor of Brightseed Bio. J.J.J.v.d.H. is a member of the Scientific Advisory Board of NAICONS Srl., Milano, Italy, and is consulting for Corteva Agriscience, Indianapolis, IN, USA. O.K. and T.S. are officers in OpenMS Inc., a non-profit foundation that manages the international coordination of OpenMS development. The other authors declare no competing interests.

## Additional information

**Correspondence and requests for materials** should be addressed to Mingxun Wang.

# Reporting Summary

## Statistics

For all statistical analyses, confirm that the following items are present in the figure legend, table legend, main text, or Methods section.

| n/a | Confirmed | |
|---|---|---|
| ☒ | ☐ | The exact sample size (*n*) for each experimental group/condition, given as a discrete number and unit of measurement |
| ☒ | ☐ | A statement on whether measurements were taken from distinct samples or whether the same sample was measured repeatedly |
| ☒ | ☐ | The statistical test(s) used AND whether they are one- or two-sided *Only common tests should be described solely by name; describe more complex techniques in the Methods section.* |
| ☒ | ☐ | A description of all covariates tested |
| ☒ | ☐ | A description of any assumptions or corrections, such as tests of normality and adjustment for multiple comparisons |
| ☒ | ☐ | A full description of the statistical parameters including central tendency (e.g. means) or other basic estimates (e.g. regression coefficient) AND variation (e.g. standard deviation) or associated estimates of uncertainty (e.g. confidence intervals) |
| ☒ | ☐ | For null hypothesis testing, the test statistic (e.g. *F*, *t*, *r*) with confidence intervals, effect sizes, degrees of freedom and *P* value noted *Give P values as exact values whenever suitable.* |
| ☒ | ☐ | For Bayesian analysis, information on the choice of priors and Markov chain Monte Carlo settings |
| ☒ | ☐ | For hierarchical and complex designs, identification of the appropriate level for tests and full reporting of outcomes |
| ☒ | ☐ | Estimates of effect sizes (e.g. Cohen's *d*, Pearson's *r*), indicating how they were calculated |

*Our web collection on statistics for biologists contains articles on many of the points above.*

## Software and code

Policy information about availability of computer code

| Data collection | Not applicable |
|---|---|
| Data analysis | All source code is publicly available here https://github.com/mwang87/MassQueryLanguage |

For manuscripts utilizing custom algorithms or software that are central to the research but not yet described in published literature, software must be made available to editors and reviewers. We strongly encourage code deposition in a community repository (e.g. GitHub). See the Nature Portfolio guidelines for submitting code & software for further information.

## Data

Policy information about availability of data

All manuscripts must include a data availability statement. This statement should provide the following information, where applicable:
- Accession codes, unique identifiers, or web links for publicly available datasets
- A description of any restrictions on data availability
- For clinical datasets or third party data, please ensure that the statement adheres to our policy

All datasets are publicly available at MassIVE (http://massive.ucsd.edu/). Accession codes for relevant datasets are included in the Supplemental Information.

# Human research participants

Policy information about studies involving human research participants and Sex and Gender in Research.

| | |
|---|---|
| Reporting on sex and gender | N/A |
| Population characteristics | N/A |
| Recruitment | N/A |
| Ethics oversight | N/A |

Note that full information on the approval of the study protocol must also be provided in the manuscript.

# Field-specific reporting

Please select the one below that is the best fit for your research. If you are not sure, read the appropriate sections before making your selection.

☒ Life sciences  ☐ Behavioural & social sciences  ☐ Ecological, evolutionary & environmental sciences

For a reference copy of the document with all sections, see nature.com/documents/nr-reporting-summary-flat.pdf

# Life sciences study design

All studies must disclose on these points even when the disclosure is negative.

| | |
|---|---|
| Sample size | N/A |
| Data exclusions | N/A |
| Replication | N/A |
| Randomization | N/A |
| Blinding | N/A |

# Reporting for specific materials, systems and methods

We require information from authors about some types of materials, experimental systems and methods used in many studies. Here, indicate whether each material, system or method listed is relevant to your study. If you are not sure if a list item applies to your research, read the appropriate section before selecting a response.

## Materials & experimental systems

| n/a | Involved in the study |
|---|---|
| ☒ ☐ | Antibodies |
| ☒ ☐ | Eukaryotic cell lines |
| ☒ ☐ | Palaeontology and archaeology |
| ☒ ☐ | Animals and other organisms |
| ☒ ☐ | Clinical data |
| ☒ ☐ | Dual use research of concern |

## Methods

| n/a | Involved in the study |
|---|---|
| ☒ ☐ | ChIP-seq |
| ☒ ☐ | Flow cytometry |
| ☒ ☐ | MRI-based neuroimaging |

