## [Peer Review File · Nature Methods]

A Universal Language for Finding Mass Spectrometry Data Patterns

Corresponding Author: Dr Mingxun Wang

A version of this paper was originally rejected for publication by Nature Methods, however that decision was reconsidered after appeal by the authors.

Version 0:

Decision Letter:

8th Aug 2023

Dear Ming,

Hope you're doing well, and apologies for the long review duration of this paper. Unfortunately one of the reviewers was unable to provide their comments after all (health reasons). Your Article entitled "A Universal Language for Finding Mass Spectrometry Data Patterns" has now been seen by 2 reviewers, whose comments are attached. While they find your work of potential interest, they have raised serious concerns which in our view are sufficiently important that they preclude publication of the work in Nature Methods, at least in its present form.

As you will see, the reviewers raise concerns about the FDR, scalability and generality outside the GNPS platform, among others.

We would be willing to look at a revised manuscript if you can fully address these criticisms (unless, of course, something similar has by then been accepted at Nature Methods or appeared elsewhere). We hope you understand that until we have read the revised paper in its entirety we cannot promise that it will be sent back for peer-review.

If you are interested in revising this manuscript for submission to Nature Methods, please contact me to discuss your appeal before making any revisions. Otherwise, we hope that you find the reviewers' comments helpful when preparing your paper for submission elsewhere.

Sincerely,
Arunima

Arunima Singh, Ph.D.
Senior Editor
Nature Methods

Reviewers' Comments:

Reviewer #1:

Remarks to the Author:

This report titled "A universal language for finding mass spectrometry data patterns" describes the development of "massQL", which is mass spec query language and is a play on words to the traditional sQL (structured query language). MassQL is designed to enable users to search public mass spectrometry data on the GNPS/MassIVE repository for specific features. From a practical perspective, more tools to search public datasets efficiently would certainly be useful.

From what I can tell, MassQL is similar to a python library to parse and extract information from .mzML files. There are many libraries to parse .mzML files. What this search is doing (matching fragments, neutral losses, etc.) is relatively trivial from a

computational perspective. There is some benefit to standardizing, but the big question is what is the comparative performance. sQL is popular not because the queries are easy to write but because sQL databases are built in a way that makes them searchable quickly, even as they get extremely large with billions of entries.

In passing, the authors mention that the compute time of searches was several hours. A crucial factor that will determine adoption of massQL into metabolomics software is the efficiency of the searches. The repository scale searches are powerful but from a practical perspective it would be helpful to know the efficiency and speed of the search in a more common setting. For example, what is the time required to perform a fragment search across 10,000 MS/MS spectra in a single .mzML file? What about across multiple .mzML files? How does this speed compare to existing libraries for parsing and searching spectra? On a related point, what approaches are implemented to make massQL scalable? Is a binary search performed? A description of the approach implemented in massQL should be included.

The report describes two applications in the main text, and then lists others in supplementary notes. Almost all of the applications are based on querying fragment ions in public MS/MS data. Another question is how different is this from the existing MASST function that is already in GNPS to query fragments from public MS/MS data? Are there searches that can be done now but couldn't before? I assume the answer is yes but it might be helpful to explain that in the text.

Many of the supplemental notes are more project teasers than complete examples of applications. They are difficult to review because so few details are provided. As an example the glucosides note is four sentences. No information is provided about what is being searched (eg, what is the pattern queried? fragments?). What are the results and conclusions? This section of supplemental notes feels more like statements of interest from researchers rather than a rigorous demonstration of an analytical pipeline.

Other than searching for fragments, what are other use cases for massQL? The example of isotope ratios and m/z differences from iron is intriguing but maybe not something that the majority of mass spectrometrists will be likely to use.

False positives are another question. When there are 1.4 billion MS/MS spectra, there are bound to be two random fragments separated by 52.91 that are not related to iron (if for no other reason this could be due to fragments from multiple precursors fragmented at the same time). I understand that the workflow is meant to generate leads, but if there are thousands of hits and 95% are false positives, then it becomes not very useful in practice. This is important to know.

The same question arises from the organophosphate ester application querying a phosphate product ion. There are thousands of compounds in metabolism that contain phosphates, many of which are known to fall off in the MS/MS pattern. The search doesn't seem nearly specific enough to find organophosphate esters in an efficient manner.

As with any public data, there is a question of quality control. The queries mostly appear to assume that single precursors are being fragmented. What happens when that is not the case? Can DIA datasets be easily excluded from searches? What measures can be taken to ensure that the searches are not providing misleading hits?

As a proof of principle, it would be useful to query a pattern where the ^{13}C peak is larger than the ^{12}C peak (could be in MS1 or MS2). This should never happen, unless there is an interference where another compound overlaps with the ^{13}C m/z value. When queried against all 1.4B spectra, this would provide a sense for how often false positive hits occur.

As a positive control for testing, it would also be useful to search a pattern that is expected to occur in every sample (eg, perhaps the presence of glucose, which should be present in most samples?).

Another potentially helpful suggestion for testing would be to organize subsets of data from MassIVE based on patterns that were reported in primary studies or patterns that were reported to be absent in the primary studies. Then the authors could search each subset for the known pattern and ensure that it is retrieved (or not retrieved) with high accuracy. Of course there may be some instances where the original papers were wrong, but I would expect these occurrences to be rare.

Reviewer #2:

Remarks to the Author:

This manuscript describes the massql package and its applications. The approach is described as a domain language to capture common query patterns in mass spectrometry data.

Although it is not described in the manuscript, the code repository suggests that the massql package uses the following Python tools:

- lark for language parsing, to convert SQL style queries to JSON;
- pymzml, pyteomics and matchms for data file parsing;
- pandas data frames for value filtering, caching and search.

The main contribution from massql appears to be the design of the language patterns.

There are many unmet demands in computational MS, driven by metabolomics and natural product research. This massql approach is a very welcome and encouraging contribution. The software package is well written, with good documentation. Search is indeed a foundational component of the field. Mass difference patterns are also fundamental to computational mass spectrometry and already included in numerous tools. Still, it is an elegant effort in massql to formalize their use.

The authors propose the value of massql as that “The resulting MassQL language provides users the flexibility and expressiveness to query simple and complex MS patterns within their data and across public data regardless of their expertise in computational MS.”

Is massql “a universal solution to mine MS data” as claimed, or is it primarily a filtering tool for GNPS?

The following questions are relevant to this evaluation:

Q1. Does massql serve users “regardless of their expertise in computational MS”?

The massql queries are well designed and easy to use. However, the issue is what users do with the result. Based on the design, the query often returns a large number of matched patterns. As shown clearly in the examples in the manuscript, ~27K matched spectra for Figure 2 and >300K matched spectra for Figure 3. The examples are on MS2 data, while the problem with MS1 data is likely to be more challenging by multiple magnitudes. To find the few true positives from so many matches, it requires significant work and usually coding if the functions are not in GNPS.

Q2. How to deal with false positives?

The authors are aware of this and discussed it in the Conclusion (false discovery rate etc.). But without actually addressing the issue, the scientific value of this tool is limited.

We tested massql on a LC-MS dataset of 100 human samples where a medication is known to be present in 38 of them. Massql MS1 search returned from 80 samples with positive matches of over 20 scans/spectra. Only two were returned as negatives. The false discovery rate here is unacceptable.

The authors showed in Figures 2 and 3 that significant followup work, not in massql per se, was involved in dealing with the false positives in MS2 data. But there is unlikely a workable solution for MS1 data based on massql as is.

Q3. How good is the scalability of massql?

The authors did not describe if the search against GNPS repository is file by file, or using existing infrastructure, e.g. indexing data or database. They acknowledged in the Conclusion, “[...] each MassQL query searching across a billion analytes requiring hours of compute time”. This statement is likely on MS2 data. Many MS2 search tools already exist. Flash entropy MS2 search is very fast (<https://doi.org/10.21203/rs.3.rs-2693233/v1>). How does massql compare to the existing tools?

The MS1 search we did in Q2 took about 2 seconds per mzML file for querying a single m/z value. When they were queried for an isotope mass delta (as Supplemental example 2.1), each file took ~20 seconds. That is, to perform a single query of one isotopic mass difference on 1 million MS1 data files may take 200 days on a single CPU core. If the scalability stated in the manuscript depends on infrastructures (pre-existing database, or expensive hardware), this should be explained.

We found no magic in the search functions in the source code of massql. It appears to depend on functions from pandas data frame. If that's the case, any developer should achieve similar performance by quick scripting.

Q4. What is the value that massql brings to MS2 search?

What's different from existing MS2 tools? If a spectrum is treated as a spectrum without considering the MS level, what's different from other MS search methods?

Can the value be demonstrated outside GNPS?

Q5. What is the value that massql brings to MS1 search?

As discussed in Q2, many false positives are expected by this approach. The MS2 data usually contain a higher ratio of useful information than MS1, because human hypotheses were often driving their collection in the first place. In untargeted MS1 data from complex samples, the scan-level search as in massql now bears no confidence. It will be too erroneous and too slow, so that we see no value in using massql to mine MS1 data.

Q6. What are the alternatives to massql?

Many functions in massql can be performed by custom scripting. In our group, when we need to search for a neutral loss in the data, we write a quick script and get it done. It is understood that massql enables many more people to do queries, given that the queries are still a long way from discoveries and their values are still debatable. In this regard, massql occupies a unique niche.

There are many tools for isotope analysis. Massql offers no perceivable advantage over existing tools.

Will massql be the tool of choice for mining mass spectrometry data (independent from GNPS)? Because the design of massql caters to SQL like queries, the benefits to developers and data scientists are not clear other than data exploration. For MS2 search, the examples like Flash entropy search are more compelling. For MS1 data search, performant examples at database scales are available. Given the lack of published examples of scientific values, we believe that mining small molecular mass spectrometry data requires significant computational expertise, more sophisticated data models than raw spectra and usually a lengthy workflow. If people need a reusable software package in such workflows, massql is unlikely to be the best choice, because search at the scan level is almost useless for MS1 and there are many reusable tools in MS2 search. Finally, we have not seen the evidence of computational performance in massql.

Q7. What is the value of massql independent from GNPS?

The examples from the authors are mostly about prefiltering for GNPS applications. As wonderful as GNPS is, it is still a small

subset of computational MS. To claim it to be “a universal solution to mine MS data”, we need to see compelling examples outside GNPS.

Q8. What is the specific value of massql in data mining?

The massql package is a nice tool for searching mass spec data. But much more work is required to get discoveries out of data mining. In all the application examples, the heavy lifting was done by other GNPS tools and the scientists.

Overall, the authors demonstrated nice applications of massql to the GNPS system. However, this design leads to a large number of false positives, which still require significant effort and computational expertise to deal with. There is no evidence on the real scalability of the software. The claim of “a universal solution to mine MS data” is not substantiated by the data.

The targeted users of massql are unclear. The proposition is that massql removes the barrier for many people to mine mass spec data without coding knowledge. In reality, it creates a situation that is hard to get out without coding. Then the design compromises its reuse by developers.

The writing of the manuscript should be improved significantly. Is massql to address the search problem or the data mining problem? What are the current practices in these areas? What is the specific gap massql aims to cover? How is massql different technically from other tools? How does massql work? What are the technical innovations beyond defining a set of vocabulary? What values were brought in by massql and what by other tools in the workflow? Without such specific information, it is difficult to understand how it contributes to scientific progress and how the field can build upon it.

** For Nature Portfolio general information and news for authors, see <http://npg.nature.com/authors>.

Version 1:

Decision Letter:

Our ref: NMETH-A52230A-Z

5th Nov 2024

Dear Mingxun,

Thank you for submitting your revised manuscript "A Universal Language for Finding Mass Spectrometry Data Patterns" (NMETH-A52230A-Z). It has now been seen by the original referees and their comments are below. The reviewers find that the paper has improved in revision, and therefore we'll be happy in principle to publish it in Nature Methods, pending minor revisions to satisfy the referees' final requests and to comply with our editorial and formatting guidelines.

TRANSPARENT PEER REVIEW

ORCID

IMPORTANT: Non-corresponding authors do not have to link their ORCID but are encouraged to do so. Please note that it will not be possible to add/modify ORCID at proof. Thus, please let your co-authors know that if they wish to have their ORCID added to the paper they must follow the procedure described in the following link prior to acceptance: <https://www.springernature.com/gp/researchers/orcid/orcid-for-nature-research>

Sincerely,
Arunima

Arunima Singh, Ph.D.
Senior Editor
Nature Methods

Reviewer #1 (Remarks to the Author):

The authors have responded to my comments in an impressively thorough fashion. The manuscript is much improved and all of my questions have been addressed. Since the first submission, MassQL has picked up a lot of traction in the community and there are a number of recent publications that use it. This is the best testimony for the value of the contribution, and those new papers themselves answer most of the questions I raised. It would be really nice to officially have MassQL out -- I recommend publication of the revised manuscript as is.

Reviewer #2 (Remarks to the Author):

The revised manuscript came with a lengthy rebuttal, which addresses few of the reviewers' concerns but confirms the main limitations of innovation in this work.

1. Both reviewers stated earlier that the search functions MassQL performs were trivial, easy to do using existing tools or scripting.

The authors acknowledge this and emphasize that "the main goal of MassQL is to lower the barrier to entry for biologists and chemists without computational skills to search MS data for patterns of interest".

2. Both reviewers stressed the critical role of computational performance.

The authors acknowledge that "query speed is not the main innovation in the MassQL reference implementation". That is after they "improved the compute speed of the reference implementation of MassQL through engineering optimizations". This, with other relevant texts, means there's no innovation in query algorithms.

The table in SI Note 1 is misleading because it is based on cached data and does not represent a real-world scenario. Neither does it include comparisons to other query approaches.

The "scalability" claim on using NextFlow is misleading because it's throwing more hardware at the problem, not based on software performance.

3. Both reviewers raised the concerns that the high number of false positives from MassQL poses a significant barrier to its practical use.

The authors have added a new section on this topic, explaining that a generic solution is not available. The example of the Mohanty paper, which is an excellent work from this group of authors, is misleading because bile acids have distinct chemical structures that are hard to confuse with other molecules.

The answer to the "thought experiment" proposed by Reviewer #1 is reported in SI Note 2, "MassQL returned 749,791 and 644,108 MS1 spectra for the "true" query and the "decoy" query, respectively. This resulted in an overall estimated FDR of 85.9%." The authors cut it down to 21.7% by expert knowledge that is not in MassQL per se. This reviewer used in the previous evaluation an example of falsely identified drugs in a human dataset by MassQL. The problem here is that MassQL by design has no noise modeling. In MS1 data, much noise is often filtered by requiring consecutive scans in the chromatography. MassQL is a brute force approach, ignoring the nature of data.

Personally, I believe it is good to acknowledge what a tool does not do. Science is better served by defining the gaps than overstating the results.

4. My opening question in the initial view was: Is massql "a universal solution to mine MS data" as claimed, or is it primarily a filtering tool for GNPS?

The authors have now retracted the former. The latter is clear now. Actually, it was how Dr. Dorrestein presented MassQL at the Metabolomics Society meeting this year, a filtering tool.

The authors cited two papers in the rebuttal to show the value of "using MassQL without GNPS tools". However, they are also coauthors on these two papers, suggesting that these were not independent use of MassQL by the community.

The Mohanty paper by this group of authors is a great example of how reanalysis of repository data led to new discoveries. Could the Mohanty paper be accomplished effectively without MassQL? Absolutely yes.

Overall, the authors responded to the reviewers' concerns with mostly negative results. This manuscript, as currently written, is like a protocol paper not a method paper. A method paper needs to prove its novelty and performance in the right scientific context. Most examples and statements here are bundled to the popularity of GNPS. While MassQL is a valuable enhancement to GNPS, its significance shall be evaluated on its standalone scientific contributions, which appear to be limited.

Version 2:

Decision Letter:

3rd Mar 2025

Dear Ming,

I am pleased to inform you that your Article, "A Universal Language for Finding Mass Spectrometry Data Patterns", has now been accepted for publication in Nature Methods. The received and accepted dates will be April 7, 2023 and March 3, 2025. This note is intended to let you know what to expect from us over the next month or so, and to let you know where to address any further questions.

Over the next few weeks, your paper will be copyedited to ensure that it conforms to Nature Methods style. Once your paper is typeset, you will receive an email with a link to choose the appropriate publishing options for your paper and our Author Services team will be in touch regarding any additional information that may be required. It is extremely important that you let us know now whether you will be difficult to contact over the next month. If this is the case, we ask that you send us the contact information (email, phone and fax) of someone who will be able to check the proofs and deal with any last-minute problems.

If you are active on Twitter/X, please e-mail me your and your coauthors' handles so that we may tag you when the paper is published.

Best regards,
Arunima

Arunima Singh, Ph.D.
Senior Editor
Nature Methods

** Visit the Springer Nature Editorial and Publishing website at www.springernature.com/editorial-and-publishing-jobs for more information about our career opportunities. If you have any questions please click here. **

Open Access This Peer Review File is licensed under a Creative Commons Attribution 4.0 International License, which permits use, sharing, adaptation, distribution and reproduction in any medium or format, as long as you give appropriate credit to the original author(s) and the source, provide a link to the Creative Commons license, and indicate if changes were made. In cases where reviewers are anonymous, credit should be given to 'Anonymous Referee' and the source. The images or other third party material in this Peer Review File are included in the article's Creative Commons license, unless indicated otherwise in a credit line to the material. If material is not included in the article's Creative Commons license and your intended use is not permitted by statutory regulation or exceeds the permitted use, you will need to obtain permission directly from the copyright holder.

Reviewer #1:

Remarks to the Author:

This report titled "A universal language for finding mass spectrometry data patterns" describes the development of "massQL", which is mass spec query language and is a play on words to the traditional sQL (structured query language). MassQL is designed to enable users to search public mass spectrometry data on the GNPS/MassIVE repository for specific features. From a practical perspective, more tools to search public datasets efficiently would certainly be useful.

Thanks for this supportive comment. We sincerely appreciate the reviewers' time to review our manuscript. The comments from both reviewers were very constructive and have significantly helped us improve our manuscript.

We very much agree that the ability to leverage entire metabolomics data repositories to make biological discoveries is underutilized. However, we would like to clarify that, although in the present manuscript MassQL is showcased on two applications of repository-scale data mining, MassQL as a language and tool ecosystem can be used on single files, within and across datasets, and it is capable of analyzing full repositories. That being said, the main goal of MassQL is to lower the barrier to entry for biologists and chemists without computational skills to search MS data for patterns of interest. We have now substantially revised the main text in order to make this clearer for the reader and our target audience (i.e., non-computational scientists).

From what I can tell, MassQL is similar to a python library to parse and extract information from .mzML files. There are many libraries to parse .mzML files. What this search is doing (matching fragments, neutral losses, etc.) is relatively trivial from a computational perspective. There is some benefit to standardizing, but the big question is what is the comparative performance. sQL is popular not because the queries are easy to write but because sQL databases are built in a way that makes them searchable quickly, even as they get extremely large with billions of entries.

This is a good point and we agree that it is possible for a computational scientist to write bespoke software to accomplish the capabilities of a single query from MassQL. However, this task may not be trivial for researchers lacking a training in computer science (i.e., the vast majority of chemists and biologists). The main goal of MassQL is to lower the barrier to entry for chemists, biologists, and mass spectrometry scientists without programming skills to search their own MS data for patterns of interest. The conciseness and expressiveness of the MassQL language, which builds upon common mass spectrometry (MS) terminology, make MassQL queries easy to understand and alter for scientists with basic familiarity with MS. This empowers non-computational scientists with the ability to leverage their domain knowledge of the chemical (or compound class) under investigation, without the need for programming skills or a dedicated computational collaborator. To further increase accessibility to the MassQL ecosystem for non-computational scientists, we have created extensive accompanying infrastructure (e.g., MassQL documentation, MassQL Compendium, MassLQ sandbox for query testing, and the new addition of a MassQL Chatbot), which altogether makes writing MassQL queries much more straightforward for non-computational scientists.

The effectiveness of MassQL in lowering the barrier to entry to search MS data for non-computational scientists is evidenced by the breadth and number of scientists utilizing it in the numerous showcase examples. Additionally, we are pleased to see that some of the showcase examples have now turned into actual publications and many other manuscripts that used MassQL to make discoveries have been published over that past year (see list below). We believe this demonstrates the utility and usability of the tool. List of publications that used MassQL:

- Discovery of new bile acids at repository scales (*Cell*)
<https://doi.org/10.1016/j.cell.2024.02.019>
- Mining of lipid changes in parasitic disease (*Nature Communications*, adapted from **SI Note 4.3**)
<https://doi.org/10.1038/s41467-023-42247-w>
- Discovery of a fungal pentose-binding biotransformation enzyme (*PNAS*)
<https://doi.org/10.1073/pnas.2301007120>
- Biomonitoring of polyphenols in urine using MassQL (*Analytical Chemistry*)
<https://doi.org/10.1021/acs.analchem.3c01393>
- Link biosynthetic gene clusters and MassQL output (*PNAS Nexus*)
<https://doi.org/10.1093/pnasnexus/pgac257>
- Mining of heteromeric adducts in public datasets using MassQL (*Analytica Chimica Acta*)
<https://doi.org/10.1016/j.aca.2022.340352>
- Mining of large plant metabolomics datasets (*GigaScience* and *Nature Scientific Data*):
<https://doi.org/10.1093/gigascience/giac124>
<https://doi.org/10.1038/s41597-024-03094-6>
- MassQL-based exploration of molecular networks for natural product discovery
<https://doi.org/10.1021/acs.jnatprod.3c00750> (*Journal of Natural Products*)
<https://doi.org/10.1021/acscentsci.3c00800> (*ACS Central Science*)
- MassQL-integrated molecular networking for studying MS/MS fragmentation patterns of beauvericin analogs (*Frontiers in Molecular Biosciences*)
<https://doi.org/10.3389/fmolb.2023.1238475>
- Screening of copper adducts (*Journal of Natural Products*)
<https://doi.org/10.1021/acs.jnatprod.4c00049>
- Discovery of cupriachelin analogs (*Frontiers in Chemistry*, originally **SI Note 4.10**)
<https://doi.org/10.3389/fchem.2023.1256962>

We sincerely thank the reviewer for this comment as it gave us the possibility to clarify these points throughout the main text (see **Introduction**, **Results** and **Discussion** section) and better tailor the manuscript to our target audience (i.e., non-computational scientists).

We also agree with the reviewer that SQL is a powerful computational language and provides enormous value for scalability. As discussed in more details in the answer to the next comment, this manuscript showcased the reference implementation of MassQL, which is equipped with a query engine not specifically optimized for speed. However, the architecture of the MassQL ecosystem implements the MassQL grammar and the query engine as two separate components. This enables the development of any computational backend, including SQL-like platforms, for

improved search speed. Notably, this paradigm of developing new query engines has already begun to be realized. In fact, the MassQL language has been adopted by popular MS analysis software (MZmine, MS-DIAL, and Bruker's Metaboscape), which developed and implemented their own query engines, while maintaining the same MassQL formal grammar. We have now clarified this aspect in the **Discussion** section (see text below):

“””

We envision the MassQL computational ecosystem will grow in adoption and capability. We recognize areas for improvement within MassQL, such as enhancing the query performance of MassQL and expanding the expressiveness of the language. We have designed and scaffolded the MassQL ecosystem to be extensible in both of these respects. We have defined a format context-free grammar for MassQL that is separate from any query engine that implements the MassQL semantics (see **Methods** section). Even though the reference implementation of the MassQL query engine is not explicitly optimized for speed (see **SI Note 1**), this architecture enables the wider community to develop new query algorithms to improve search speed, while using the same formal grammar. This paradigm has already been demonstrated in the popular software tools that implemented their own query engine: MS-DIAL¹², Mzmine¹⁰, and Bruker's MetaboScape. Finally, MassQL derives its strength from a vibrant user open-source community and we expect this community-guided evolution of the language to continue in the future. As both the language and software ecosystem evolve, MassQL will become more capable and versatile to meet the scientific community's growing needs in mining MS data.

“””

In passing, the authors mention that the compute time of searches was several hours. A crucial factor that will determine adoption of massQL into metabolomics software is the efficiency of the searches. The repository scale searches are powerful but from a practical perspective it would be helpful to know the efficiency and speed of the search in a more common setting. For example, what is the time required to perform a fragment search across 10,000 MS/MS spectra in a single .mzML file? What about across multiple .mzML files? How does this speed compare to existing libraries for parsing and searching spectra? On a related point, what approaches are implemented to make massQL scalable? Is a binary search performed? A description of the approach implemented in massQL should be included.

This is an important comment and we agree with the reviewer that including more details of the practicality of the existing MassQL computational ecosystem is important for the reader. Following the reviewer's suggestion, we have performed a performance evaluation of the reference implementation of MassQL and reported the results in **SI Note 1 - MassQL query speed evaluation** as a guide to readers:

“””**SI Note 1 - MassQL query speed evaluation**

We carried out a MassQL performance evaluation using the reference implementation (version 0.0.15) on a standardized reference server in Amazon Web Services t3.large (Intel Xeon Platinum 8000 series processor (Skylake-SP or Cascade Lake)), 2 vCPU, 8 GB RAM, 20TB SSD storage - gp3 - 3000 IOPS 125MB/sec. We tested 5 different queries on a single mzML file containing 6827 spectra (with 3809 MS/MS) from Thermo Fisher Q-Exactive mass spectrometer

(https://massive.ucsd.edu/ProteoSAFe/DownloadResultFile?forceDownload=true&file=f.MSV000086206/ccms_peak/raw/S_N1.mzML). We divided queries into two categories: simple (n=4) and complex (n=1). Simple queries searched for 1-2 peaks (or neutral losses) in MS/MS spectra. The complex query searched for a mass delta pattern (+164.9, +329.8 Da) between any 3 peaks in the MS/MS spectra. These queries and corresponding measured query times (average across 10 trials) using a pre-processed cache version of the spectra are reported in **SI Table 1**.

Simple queries were run in between 0.2 and 0.3 seconds, whereas the complex query took 148 seconds to complete. “Simple” broadly means searching for specific masses or neutral losses and not using variables (e.g., patterns between any possible m/z (s) in the spectrum), which are a complex language feature. These complex language features are expected to increase query times on the reference implementation as this reference has not been optimized specifically to accelerate these complex queries.

New optimized query engines can be developed independently and use the MassQL grammar, and potentially improve query speed both in terms of computing efficiency. For example, the MassQL language has already been implemented in various MS data analysis software (MS-DIAL, MZmine, and Bruker’s MetaboScape), which use their own query engine, while keeping the same MassQL formal grammar.

Query	Query type	Query time (s)
QUERY scaninfo(MS2DATA) WHERE MS2PROD=123.0417:TOLERANCEPPM=10	Simple	0.28 ± 0.07
QUERY scaninfo(MS2DATA) WHERE MS2PROD=123.0417:TOLERANCEPPM=10 AND MS2PROD=121.0624:TOLERANCEPPM=10	Simple	0.30 ± 0.04
QUERY MS2DATA WHERE MS2PROD=226.18 AND MS2PREC=226.1797	Simple	0.20 ± 0.03
QUERY scannum(MS2DATA) WHERE MS2NL=163	Simple	0.22 ± 0.04
QUERY scaninfo(MS2DATA) WHERE MS2PROD=X:TOLERANCEMZ=0.1:INTENSITYPERCENT=5 AND MS2PROD=X+164.9:TOLERANCEMZ=0.1:INTENSITYPERCENT=5 AND MS2PROD=X+329.8:TOLERANCEMZ=0.1:INTENSITYPERCENT=5	Complex	148.82 ± 2.35

SI Table 1 - Summary of MassQL query speeds. This table demonstrates the MassQL query speed over an evaluation LC-MS/MS file from a Q-Exactive mass spectrometer (mzspec:MSV000086206:ccms_peak/raw/S_N1.mzML). Queries are divided into “simple” and “complex” queries. Simple queries search for one or two peaks (or neutral losses) in MS/MS spectra. The complex query searches for a mass delta pattern between any three peaks in the MS/MS spectra. All run times are performed on an AWS t3.large virtual machine. Measurement of query time uses a pre-processed cached version of the spectra to avoid expensive initial processing of the mzML files.

“””

Since the original submission, we have improved the compute speed of the reference implementation of MassQL through engineering optimizations. For instance, in the MassQL reference implementation, a significant portion of the compute time (>98% in some queries) was simply reading the mass spectrometry data rather than actually computing the queries. We have also included a NextFlow workflow that enables querying of multiple files in parallel on a computational cluster. The compute requirements are expected to scale linearly to the number of MS data experimental files that are provided.

With that being said, we acknowledge that query speed is not the main innovation in the MassQL reference implementation and there are opportunities to further optimize performance with strategies such as indexing or binary search. The reference implementation of MassQL is primarily geared as a point of reference for semantic correctness for MassQL queries and provides a guide for future implementations that may optimize speed or compatibility in other systems. In fact, since the grammar and query engine are two separate components in the MassQL ecosystem, new query engines can be developed for improved search speed (e.g., for specific classes of queries), while using the same formal MassQL grammar. As mentioned in the previous comment, this has already happened for the popular software packages MS-DIAL, MZmine and Bruker's MetaboScape, which developed and implemented their own query engine for improved search speed. We now clarify this aspect in both **SI Note 1** (see above) and the **Discussion** section in the main text:

“”

We envision the MassQL computational ecosystem will grow in adoption and capability. We recognize areas for improvement within MassQL, such as enhancing the query performance of MassQL and expanding the expressiveness of the language. We have designed and scaffolded the MassQL ecosystem to be extensible in both of these respects. We have defined a format context-free grammar for MassQL that is separate from any query engine that implements the MassQL semantics (see **Methods** section). Even though the reference implementation of the MassQL query engine is not explicitly optimized for speed (see **SI Note 1**), this architecture enables the wider community to develop new query algorithms to improve search speed, while using the same formal grammar. This paradigm has already been demonstrated in the popular software tools that implemented their own query engine: MS-DIAL¹², Mzmine¹⁰, and Bruker's MetaboScape. Finally, MassQL derives its strength from a vibrant user open-source community and we expect this community-guided evolution of the language to continue in the future. As both the language and software ecosystem evolve, MassQL will become more capable and versatile to meet the scientific community's growing needs in mining MS data.

“”

Furthermore, we have added a **Methods** section that describes the various components of the MassQL reference implementation:

“”

MassQL reference implementation

- The reference implementation is a fully working version of the MassQL software ecosystem for the community to use. It also serves as a guide for future MassQL implementations that may optimize speed and/or introduce new functions in other systems. Specifically, the reference implementation includes the following pieces:
 - MassQL formal grammar. The grammar is defined using the extended backus-naur form and builds upon common MS terminology for improved expressiveness and interpretability (see **MassQL Language Description** section). During the development, community input and feedback shaped the vocabulary and capabilities of the language.
 - MassQL parser. The parser transforms a query into an internal data structure that can be used by any programming language. The parsing is done by using the lark Python library (<https://github.com/lark-parser/lark>) and specific Python code to transform a MassQL query to a parse tree and into the internal data structure that organizes all query conditions and qualifiers.
 - MassQL query engine. The MassQL reference query engine is written in Python and utilizes pyteomics⁴⁵ to read open mass spectrometry data files from mzML, mzXML, and MGF formats into data frames. Such MS spectra in data frame format can optionally be saved as Apache feather files to cache data for repeated querying. The query engine itself processes the query over these data frames using the Python pandas library to perform data filtering and manipulations. Output results are data frames that can be exported as a tabular format. Optionally, the retrieved MS spectra can be exported in JSON format, MGF, and mzML⁴⁶.
- MassQL NextFlow workflow. The MassQL reference NextFlow⁴⁷ workflow is designed as an automated high-throughput tool for querying multiple files simultaneously on a computational cluster. This workflow utilizes the reference query engine and parallelizes the querying of multiple files across a multi-core processor or a batch cluster, depending on the compute resources available. All results are then merged together, including extracted MS spectra in JSON format, MGF, and mzML format, if desired.

“””

With that being said, we would like to point out that, despite not being explicitly aimed for speed, the reference implementation of MassQL has already proven practically usable for discovery tasks. For example, for the organophosphate example in the main text, the repository-scale query used a total of **810.1** CPU-hours (~12 wall-hours on a modern desktop workstation), which included the inefficient parsing of mzML files. We expect the updated reference implementation to be significantly faster using pre-processed data, which is tractable on a modern commodity workstation with 32-64 CPU cores.

We again thank the reviewer for their comments as it gave us the possibility to clarify these aspects throughout the manuscript.

The report describes two applications in the main text, and then lists others in supplementary notes. Almost all of the applications are based on querying fragment ions in public MS/MS data. Another question is how different is this from the existing MASST function that is already in GNPS

to query fragments from public MS/MS data? Are there searches that can be done now but couldn't before? I assume the answer is yes but it might be helpful to explain that in the text.

This is a very important point and a very similar question was asked by Reviewer #2. We sincerely thank both reviewers for raising this point as it gave us the possibility to clarify this aspect in the manuscript.

MassQL and MS/MS search tools (e.g., spectral library match, MASST) are fundamentally different. MS/MS search tools are constrained to utilize the full MS/MS spectrum to retrieve a match - i.e., a match is retrieved when the overall MS/MS similarity is above a user-defined threshold. In contrast, MassQL queries can be seen as a set of user-defined conditions that need to be satisfied for a spectrum to be retrieved. This provides the user with the flexibility to search for more specific and complex patterns (e.g., combine MS1 and MS/MS patterns, include retention and drift time constraints) and empowers scientists to leverage their domain knowledge of the chemical (or compound class) under investigation. For example, it is known that small structural modifications can result in large changes in the overall fragmentation patterns, causing potentially relevant molecules to evade discovery by similarity-based search. This can be avoided by designing MassQL queries that specifically search for the presence (or absence) of key fragments or neutral losses in the MS/MS spectrum, regardless of the overall similarity. As an example, Selegato et al. shows how MassQL could retrieve spectra of beauvericins analogs that could not be discovered by Feature-Based Molecular Networking (an approach that utilizes full MS/MS similarity) alone due to large changes in fragmentation patterns (<https://doi.org/10.3389/fmolb.2023.1238475>). In addition, information about the precursor ion information, retention, and drift time, as well as any combination thereof can be included as additional query constraints.

Thus, MassQL is complementary to existing MS/MS matching tools as it gives scientists more flexibility to encode their domain-specific knowledge into a MassQL query. We have now clarified this aspect in the **Discussion** section:

“”

While traditional MS/MS similarity/search tools are a powerful technique for matching related or similar compounds within libraries or repositories (e.g., spectral library search⁴³, MASST⁴⁴), MassQL provides a complementary set of capabilities with enhanced flexibility and precision. Specifically, traditional MS/MS search tools rely on a full MS/MS similarity measure to retrieve a match, whereas MassQL queries are a set of user-defined constraints to retrieve a spectrum. This provides the flexibility to search for more specific and complex patterns (e.g., combine MS1 and MS/MS patterns, retention and drift time constraints, see **SI Note 3.3 and 3.4**) and empowers scientists to leverage their domain knowledge of the chemical or compound class under investigation. A specific example where MassQL can complement MS/MS similarity is when small structural modifications can result in large changes in the overall fragmentation patterns. This situation can cause relevant analog molecules to evade discovery by MS/MS similarity-based search. MassQL has been shown to complement MS/MS comparison strategies by enabling the

searching for conserved key fragments or neutral losses in the MS/MS spectrum without requiring a full MS/MS similarity match (for example in Selegato et al.⁴²).

“”

Many of the supplemental notes are more project teasers than complete examples of applications. They are difficult to review because so few details are provided. As an example the glucosides note is four sentences. No information is provided about what is being searched (eg, what is the pattern queried? fragments?). What are the results and conclusions? This section of supplemental notes feels more like statements of interest from researchers rather than a rigorous demonstration of an analytical pipeline.

We thank the reviewer for this comment. Our intention with the large number of use cases in the Supplementary material was to showcase how scientists with different backgrounds could successfully develop a wide variety of MassQL queries based on their domain-specific knowledge, thereby highlighting the tool's usability. However, we agree with the reviewer that this might be difficult to review, navigate, and understand for both the reviewer and the reader.

We have now reorganized the application examples in the SI as follows:

- **SI Note 3 - Highlighted MassQL Applications** contains the most detailed applications, selected in order to showcase the different technical elements in MassQL - i.e., search for MS2 product ions, search for MS2 product ions + neutral losses, search for MS1 patterns, search for MS1 + MS2 patterns, inclusion of ion mobility constraints, etc.
- **SI Note 4 - Extended MassQL applications examples.** This section now includes all others application examples. While some examples may be redundant in terms of MassQL language features being utilized, our goal is to demonstrate the breadth of research contexts in which MassQL can be applied. Given the target audience of this manuscript (i.e., non-computational scientists), we believe these application examples can serve as inspiration and/or starting point for query development, even if lacking extensive details and review.

We hope this new organization will improve readability and navigation of the manuscript.

We additionally thank the reviewer for pointing out potential difficulties in understanding the specific data patterns searched for in each application example. To improve readability and interpretability, we have added a “**MassQL Translation**” box in each application example, which contains an automated machine translation of the MassQL queries. This will facilitate interpretation of the query for new users unfamiliar with the MassQL syntax. For example, in the case of the glucoside application mentioned by the reviewer, the MassQL Translation box will be:

“”

MassQL Query

For O-hexoglycosidase

QUERY scaninfo(MS2DATA) WHERE MS2PROD=162.05 (for galactose or glucose)

QUERY scaninfo(MS2DATA) WHERE MS2PROD=176.04 (for glucuronic acid)

For C-hexglycosides

QUERY scaninfo(MS2DATA) WHERE MS2PROD=162.05 (glucose)

QUERY scaninfo(MS2DATA) WHERE MS2PROD=120.04 (glucose fragmentation)

QUERY scaninfo(MS2DATA) WHERE MS2PROD=90.04 (glucose fragmentation)

For O-pentoglycosides

QUERY scaninfo(MS2DATA) WHERE MS2PROD=132.06 (arabinose)

QUERY scaninfo(MS2DATA) WHERE MS2PROD=145.06 (rhamnose)

MassQL Translation

Returning the scan information on MS2.

The following conditions are applied to find scans in the mass spec data.

Finding MS2 neutral loss peak at m/z 162.05

Returning the scan information on MS2.

The following conditions are applied to find scans in the mass spec data.

Finding MS2 neutral loss peak at m/z 176.04

Returning the scan information on MS2.

The following conditions are applied to find scans in the mass spec data.

Finding MS2 neutral loss peak at m/z 162.05

Finding MS2 neutral loss peak at m/z 120.04

Finding MS2 neutral loss peak at m/z 90.04

Returning the scan information on MS2.

The following conditions are applied to find scans in the mass spec data.

Finding MS2 neutral loss peak at m/z 132.06

Finding MS2 neutral loss peak at m/z 145.06

“””

With regard to readability and interpretability of MassQL queries, we have also developed and deployed a large language model (LLM)-powered conversational assistant to support new users with real-time MassQL query writing and troubleshooting. The MassQL conversational assistant is now described in the **Results** section in the manuscript and can be accessed at: https://massql-analysis.gnps2.org/MassQL_Chatbot. A typical output from the MassQL Chatbot for the glucoside note mentioned above can be viewed here:

https://massql-analysis.gnps2.org/MassQL_ChatHistory?chat_id=83e5c740-9e0c-4a76-8609-d2178c78c9ed

Other than searching for fragments, what are other use cases for massQL? The example of isotope ratios and m/z differences from iron is intriguing but maybe not something that the majority of mass spectrometrists will be likely to use.

We again thank the reviewer for this question as this speaks to both a lack of clarity about the MassQL capabilities in the main text, as well as a confusing organization of the Supplementary Information.

Besides searching for specific fragments in MS/MS fragmentation spectra, MassQL enables:

- Search, or exclude, patterns in MS1 data (e.g., isotopic patterns, adduct mass shift)
- Exclude fragments in MS/MS fragmentation spectra (using the EXCLUDE condition)
- Search, or exclude, neutral losses in MS/MS fragmentation spectra
- Apply chromatographic and ion mobility constraints

Moreover, each of these query elements can be combined with Boolean operators (e.g. AND, OR) to form more complex queries. An illustrative example is provided in **SI Note 3.3.4**, where a pattern in the MS1 is combined with a pattern in the MS/MS.

As mentioned in the previous comment, we have now reorganized the application examples in the Supplementary material so that “**SI Note 3 - Highlighted MassQL Applications**” contains selected use cases that best showcase these technical elements in MassQL. We hope this reorganization will improve readability of the manuscript and highlight the different language functions.

Concerning the siderophores example, we are happy to hear that the reviewer finds it intriguing and we believe MassQL will be useful for this subfield, even if not used by the majority of mass spectrometrists. The goal of MassQL is to provide the wider audience of non-computational scientists with the ability and flexibility to search for data patterns in their own MS data, regardless of their research field or application. To perform the same searches, before MassQL, the same scientists were required to either possess the necessary programming skills to write bespoke scripts, or to have access to a dedicated computational collaborator. We believe this will have a great impact especially in niche research applications, which are less supported by the computational MS community and for which dedicated tools do not exist (or are no longer maintained). The query for iron-binding molecules, as well as many of the application examples in the Supplementary material, are examples of small niches where the availability of a tool like MassQL may be crucial. This has already been realized with the iron isotopic queries, with scientists quickly adapting the iron queries in this manuscript to search for copper adducts in their own data – and have already demonstrated this value in a publication ([10.1021/acs.jnatprod.4c00049](https://doi.org/10.1021/acs.jnatprod.4c00049)).

False positives are another question. When there are 1.4 billion MS/MS spectra, there are bound to be two random fragments separated by 52.91 that are not related to iron (if for no other reason this could be due to fragments from multiple precursors fragmented at the same time). I understand that the workflow is meant to generate leads, but if there are thousands of hits and 95% are false positives, then it becomes not very useful in practice. This is important to know.

This is a very good point, and a very similar comment was raised also by Reviewer #2.

Overall we believe that addressing the issue of FDR in MassQL is not an easy task and an universal method for FDR estimation cannot be established due to the nature of MassQL itself. Being a flexible query language, the FDR depends on the nature of the data being queried, the query itself, and the expectation of what is in the result. Very specific and well-designed queries will result in lower FDR, even when used at a repository scale. We believe that tailored results

validation strategies should be designed by the user, case-by-case depending on the goal of the study. This has been showcased by Mohanty et al. in a recent publication that used MassQL to mine public LC-MS/MS data at a repository scale and discover new, unreported bile acids (Mohanty et al. 2024, <https://doi.org/10.1016/j.cell.2024.02.019>). In this publication, the authors developed MassQL queries for retrieving bile acids of MS/MS spectra from the public MassIVE/GNPS data repository. Before the actual repository-scale search, the authors estimated the FDR of the designed MassQL queries using the GNPS MS/MS spectral library as a reference. The GNPS MS/MS spectral library contains MS/MS spectra associated with known structural annotations (4,533 of which belong to bile acids). Therefore, the MS/MS spectra retrieved by each MassQL query could be compared to the MS/MS spectra expected to be retrieved. Mohanty et al. assessed the selectivity of each MassQL query by counting the number of non-bile acid MS/MS spectra “wrongly” retrieved from the spectral library. By doing so, the authors could iteratively refine their queries until a suitable FDR was reached (estimated at ~0.27%) before the actual repository-scale search. The actual repository-scale search retrieved more than 270K putative bile acid MS/MS spectra from the MassIVE/GNPS repository.

It must be noted that the same FDR estimation approach described above cannot be used when performing searches at the MS1 level. As this reviewer suggests in a comment below, a potential strategy for assessing the FDR of MS1-level searches is to use a “decoy” query that searches for a ¹³C isotopic pattern that is very unlikely to occur (i.e., ¹³C isotope peak in the MS1 spectra larger than the ¹²C isotope peak). We have addressed this in more detail in the answer to the corresponding reviewer’s comment (see below).

We sincerely thank both reviewers for bringing this up as this is something that can generate confusion and we should have better discussed in the main text. We now added a “**False Discovery Rate estimation and query validation strategies**” section in the **Results** of the manuscript.

“””

False Discovery Rate estimation and query validation strategies

Overall, the key challenge when using MassQL is to define queries that are sensitive towards the target compound(s) (i.e., effectively retrieve the desired spectra), but do not retrieve too many false positive hits. Due to the flexibility and broad applicability of MassQL, a universal method for false discovery rate (FDR) estimation is difficult to establish. Rather, tailored strategies for query validation can be designed case by case by the user, depending on the research context.

As demonstrated in a recent publication that used MassQL to mine LC-MS/MS raw data in the public domain and discover new, unreported bile acids³¹, one possible strategy to estimate the FDR of MassQL queries over MS/MS spectra, is to putatively identify the MS/MS spectra retrieved by MassQL using reference MS/MS spectral libraries. In their study, Mohanty et al. first used the GNPS spectral libraries (which contained 4,533 reference spectra of bile acids) to design and refine MassQL queries for bile acid spectra and estimate the queries’ selectivity. Such selectivity was measured by counting the number of retrieved bile acids and the number of retrieved non-bile acids (false positives). Thereafter, when using the refined MassQL queries to search repository data, more than 594K putative bile acids MS/MS spectra were retrieved. Of

these, 270,437 MS/MS spectra were putatively identified by MS/MS library search, with 726 MS/MS matching to non-bile acids (0.27%). It is important to highlight that such FDR estimation approach is limited to the compounds that are deposited in the reference MS/MS libraries.

It should be noted that the same validation approach cannot be universally applied, for example, to the repository-scale discovery of siderophores described in the present manuscript (see **Results - Discovery of siderophores at repository scale**). Siderophore molecules do not belong to a single compound class and can exhibit very diverse chemical structures. Therefore, queries for diagnostic MS/MS fragment ions cannot be used and the search was instead performed on the MS1 level (see **Results - Discovery of siderophores at repository scale**). The adopted strategy was to first develop and refine the query on a reference dataset of *Eutypa lata* extracts known to contain iron-binding molecules¹⁷. After satisfactory results were obtained on the reference dataset (**SI Note 3.3.2**), the query was performed on a repository scale. In this specific example, we used a strategy based on a “decoy” query to get a sense for potential false discoveries (**SI Note 2**) - estimated at 21.7% for this repository scale query. Although a relatively large number of false positive hits may be expected when performing searches at a repository scale, these false positives can be mitigated by utilizing additional computational tools in a downstream analysis (e.g., molecular networking, spectral library search) to validate the query results. In these cases, MassQL acts more as a pre-filter to reduce the data to a more tractable size (from hundreds of millions of spectra to a few thousands) and “enrich” them with putative leads for further investigation and confirmatory experiments (see **Results - Discovery of siderophores at repository scale** and **Organophosphate esters in the environments**). Overall, we encourage users to critically inspect query results and develop tailored validation strategies that are fit for their research purposes.

“”

The same question arises from the organophosphate ester application querying a phosphate product ion. There are thousands of compounds in metabolism that contain phosphates, many of which are known to fall off in the MS/MS pattern. The search doesn't seem nearly specific enough to find organophosphate esters in an efficient manner.

The reviewer is absolutely correct that phosphate groups are common in biology and, consequently, a large number of false positive hits may be expected. We believe that in these situations (i.e., expected large number of false positives) MassQL can be efficiently combined with downstream complementary analysis tools to mitigate this problem and assist the validation of the MassQL results. In the organophosphate ester application, we showcase how molecular networking can be used to drive the linkage of molecular families of interest (i.e., organophosphate esters), while spectral library search can be used to annotate the molecular network and assist its exploration. A similar approach was used also in the repository-scale search of siderophores (see **Results - Discovery of siderophores at repository scale** section). While in principle one could build an entire molecular network of all public data and find the molecular family that includes the organophosphate ester, this approach would be computationally prohibitive. In fact, this would require creating a molecular network of nearly 700 times more data, in a process that is quadratic in time complexity. Therefore, in these applications, MassQL can be seen as a pre-filter to reduce the data to a more tractable size (from hundreds of

millions of spectra to a few thousands) and “enrich” them with good leads for further investigation and confirmatory experiments.

These are very important aspects and we again thank the reviewer for bringing them up. In the updated version of the manuscript, we cover these aspects in both the “**False Discovery Rate estimation and query validation strategies**” section:

“”

Although a relatively large number of false positive hits may be expected when performing searches at a repository scale, these false positives can be mitigated by utilizing additional computational tools in a downstream analysis (e.g., molecular networking, spectral library search) to validate the query results. In these cases, MassQL acts more as a pre-filter to reduce the data to a more tractable size (from hundreds of millions of spectra to a few thousands) and “enrich” them with good leads for further investigation and confirmatory experiments (see **Results - Discovery of siderophores at repository scale and Organophosphate esters in the environments**).

“”

And **Discussion** section:

“”

First, in this manuscript, we showcased how discovery can be enhanced by combining upstream MassQL searches with downstream molecular networking and spectral library analysis (see **Results - Discovery of siderophores at repository scale and Organophosphate esters in the environments** sections). The use of MassQL as a pre-filtering tool was essential in making the analysis possible, both from a computational tractability and data interpretability/prioritization perspective. Second, MassQL has been shown in the literature as a downstream tool to enhance the analysis of molecular networking - specifically to aid in the prioritization of relevant compounds and, as highlighted above, to overcome shortcomings in MS/MS alignment^{35,37-39,41,42,42}.

“”

As with any public data, there is a question of quality control. The queries mostly appear to assume that single precursors are being fragmented. What happens when that is not the case? Can DIA datasets be easily excluded from searches? What measures can be taken to ensure that the searches are not providing misleading hits?

We thank the reviewer for this important question. Chimeric spectra are indeed common in LC-MS/MS-based metabolomics and lipidomics. MassQL provides the flexibility to handle and/or mitigate issues arising from chimeric spectra in different ways, depending on the research question and context. Some investigators might be interested in retrieving all spectra, including chimeric ones, while others might want to retrieve only “pure” spectra.

Chimeric spectra are challenging to work with for MS/MS-similarity-based tools such as traditional MS/MS search tools (e.g., spectral library match or MASST) and molecular networking. These tools rely on the overall MS/MS similarity to retrieve a match or connect two nodes in a molecular network. MS/MS similarity is heavily affected (i.e., lowered) by the presence of “contaminant”

peaks in the MS/MS spectrum when more than one ion is co-fragmented. There are scenarios where the researcher might be interested in retrieving these chimeric spectra, regardless of the (lowered) MS/MS similarity. In these situations, MassQL can be used to formulate queries that specifically search for the presence of MS/MS fragments that are highly diagnostic for the chemical or compound class of interest (see, for example, use cases in **SI Note 3.2 - MassQL Query using MS/MS Spectral Information**). MassQL queries can be seen as a set of conditions that need to be satisfied in a spectrum to retrieve a match, independent of overall MS/MS similarity. We now discuss this aspect in more details in the **Discussion** section:

“””

While traditional MS/MS similarity/search tools are a powerful technique for matching related or similar compounds within libraries or repositories (e.g., spectral library search⁴³, MASST⁴⁴), MassQL provides a complementary set of capabilities with enhanced flexibility and precision. Specifically, traditional MS/MS search tools rely on a full MS/MS similarity measure to retrieve a match, whereas MassQL queries are a set of user-defined constraints to retrieve a spectrum. This provides the flexibility to search for more specific and complex patterns (e.g., combine MS1 and MS/MS patterns, retention and drift time constraints, see **SI Note 3.3 and 3.4**) and empowers scientists to leverage their domain knowledge of the chemical or compound class under investigation. A specific example where MassQL can complement MS/MS similarity is when small structural modifications can result in large changes in the overall fragmentation patterns. This situation can cause relevant analog molecules to evade discovery by MS/MS similarity-based search. MassQL has been shown to complement MS/MS comparison strategies by enabling the searching for conserved key fragments or neutral losses in the MS/MS spectrum without requiring a full MS/MS similarity match (for example in Selegato et al.⁴²).

“””

On the other hand, there are scenarios where the researcher might be interested in discarding chimeric spectra from the MassQL search. In these situations, various strategies can be used. For example, one could use the INTENSITYPERCENT filter (e.g., MS2PROD=163.1:INTENSITYPERCENT=10) to discard spectra where the relative intensity of diagnostic MS/MS peak(s) is lower than a certain threshold. In fact, in “highly contaminated” chimeric MS/MS spectra, the relative intensity of a diagnostic fragment will be proportionally lower compared to the corresponding non-chimeric spectra. This strategy is, for example, routinely applied by one of the co-authors of the present manuscript (i.e., Laura-Isobel McCall) when searching for glycerophosphocholine MS/MS spectra. Alternatively, one could use the EXCLUDE condition (e.g., MS2PROD=163.1:EXCLUDED) to explicitly discard all spectra containing a specific MS/MS peak that should never be present. This could be the case for MS/MS peaks known to belong to a particular and/or well-known contaminant, or for MS/MS peaks that are known to be impossible to arise from the fragmentation of the chemical (or compound class) under investigation.

We would like to highlight that in all these situations, domain-specific knowledge of the chemical or compound class of interest is required, and this is primarily brought by chemists and biologists

in their field. We believe that MassQL now empowers these scientists to easily encode their domain knowledge into concise and expressive MassQL queries.

Finally, regarding the exclusion of DIA datasets from the search, a possibility in GNPS is to restrict the search to data acquired on Thermo Q Exactive data, which are for the vast majority acquired in DDA mode. This is, for example, the strategy that we adopted in the application examples highlighted in the main text of this manuscript. Another possible option is to exclude all MS/MS spectra that are assigned to an “artificial” precursor m/z .

For example, in Waters MSe data (a common DIA strategy), all MS/MS spectra are assigned to an “artificial” precursor m/z , normally placed in the middle of the MS1 scan range - see, for example the straight blue line of MS/MS scans at m/z 770.0 in the Data Exploration panel of the GNPS dashboard (example data Link). In these situations, two approaches can be used (and added to the MassQL query using Boolean conjunctions):

- 1) Explicitly exclude MS/MS spectra assigned to the “artificial” precursor m/z :

```
QUERY MS2DATA WHERE MS2PREC=770.0:TOLERANCEPPM=1:EXCLUDED
```

- 2) Filter likely “artificial” precursor m/z , based on mass defect

```
QUERY MS2DATA WHERE MS2PREC=ANY:MASSDEFECT=massdefect(min=0, max=0.00001):EXCLUDED
```

As a proof of principle, it would be useful to query a pattern where the ^{13}C peak is larger than the ^{12}C peak (could be in MS1 or MS2). This should never happen, unless there is an interference where another compound overlaps with the ^{13}C m/z value. When queried against all 1.4B spectra, this would provide a sense for how often false positive hits occur.

That is another interesting suggestion and we thought a lot about it. As the reviewer suggested, we used the repository-scale search of siderophores (described in the main text) as a representative example and created a “decoy” query that includes an isotopic pattern that is very unlikely to occur for singly charged organic compounds below 1000 Da - i.e., ^{13}C isotope peak larger than the ^{12}C isotope peak in the MS1 spectrum. A comparison of the results of the “true” vs “decoy” queries is now described in **“SI Note 2 - False Discovery Rate estimation of siderophores at repository scale”** (corresponding text reported below). Although we agree with the reviewer that this can provide a sense of FDR at the MS1 level, we note that these results can hardly be generalized to all MS1 queries. In fact, as discussed in a comment above, more specific and well-designed queries will result in lower FDR, even when used at a repository scale.

“”SI Note 2 - False Discovery Rate estimation of siderophores at repository scale

In the main manuscript, we discuss a possible strategy to assess FDR of MassQL searches at the MS/MS level, as demonstrated by Mohanty et al.¹ (see **False Discovery Rate estimation and query validation strategies**). However, the same FDR estimation strategy can not be used when performing searches at the MS1 level. A potential strategy for assessing the FDR of an MS1 query is to create a “decoy” query that includes isotopic patterns that are unlikely to occur for the target chemical or compound class under investigation. In the specific case of the siderophore query, we created a “decoy” query that includes an unlikely ¹²C/¹³C isotopic profile - i.e., ¹³C isotope peak larger than the ¹²C isotope peak in the MS1 spectrum (**SI Figure 2.1**). This is very unlikely to occur for singly charged organic compounds below 1500 Da, especially for natural products. We ran the following “true” and “decoy” MassQL queries on all public high-resolution Thermo Fisher Q Exactive data available in the GNPS/MassIVE repository and compared the results.

“True” MassQL Query:

```
QUERY scaninfo(MS1DATA) WHERE MS1MZ=X-
1.993:INTENSITYMATCH=Y*0.063:INTENSITYMATCHPERCENT=25:TOLERANCEPPM=10 AND
MS1MZ=X:INTENSITYMATCH=Y:INTENSITYMATCHREFERENCE:INTENSITYPERCENT=5 AND
MS1MZ=X+1:INTENSITYMATCH=Y*0.5:INTENSITYMATCHPERCENT=60 AND MS1MZ=X-
52.91:TOLERANCEPPM=10 AND MS2PREC=X-52.91
```

“Decoy” MassQL Query:

```
QUERY scaninfo(MS1DATA) WHERE MS1MZ=X-
1.993:INTENSITYMATCH=Y*0.063:INTENSITYMATCHPERCENT=25:TOLERANCEPPM=10 AND
MS1MZ=X:INTENSITYMATCH=Y:INTENSITYMATCHREFERENCE:INTENSITYPERCENT=5 AND
MS1MZ=X+1:INTENSITYMATCH=Y*2.0:INTENSITYMATCHPERCENT=60 AND MS1MZ=X-
52.91:TOLERANCEPPM=10 AND MS2PREC=X-52.91
```

SI Figure 2.1 - Graphical representation of the “true” (A) and “decoy” (B) MassQL queries. Green peaks represent the MS1 pattern being searched by the query. In the “decoy” query, a ¹³C isotope peak larger than the corresponding ¹²C is being searched. Plots were created using the MassQL sandbox (<https://massql.gnps2.org/>).

MassQL returned 749,791 and 644,108 MS1 spectra for the “true” query and the “decoy” query, respectively. This resulted in an overall estimated FDR of 85.9%. However, the *m/z* distribution of the returned MS1 hits by both queries are heavily skewed towards low *m/z* values (**SI Figure 2.2A**). Since microbial siderophores reported in literature predominantly lie in the mass range of

500-1500 Da², we restricted the count to this m/z range (**SI Figure 2.2B**). By doing so, the number of MS1 matches are 107,445 and 23,421 for the “true” and “decoy” query, respectively. This results in an estimated FDR of 21.7%.

SI Figure 2.2 - Distribution of m/z associated with the MS1 hits returned by the “true” and “decoy” MassQL queries (histogram bins of 1Da). Each returned MS1 spectrum is associated with the m/z matching the specific query pattern (i.e., X variable in the MassQL query). Distribution of m/z for the “true” and “decoy” query are displayed in blue and orange, respectively. A) Results considering the full 100-1500 m/z range (FDR=85.9%); B) Results considering the 500-1500 m/z range.

As a positive control for testing, it would also be useful to search a pattern that is expected to occur in every sample (eg, perhaps the presence of glucose, which should be present in most samples?).

We thank the reviewer for this comment and we really like the spirit of this question. Creating universally-applicable positive control queries as suggested by the reviewer might be difficult due to the extreme heterogeneity of the sample types each user might be dealing with. Nonetheless, we agree with the reviewer that it is critical for both developers and users to ensure the ability of MassQL queries to retrieve the target molecules and/or MS data patterns they are intended for.

From a tool development perspective, we employed a test-driven methodology during the development of the MassQL query engine and formal grammar. That is, we used public datasets known to contain specific MS data patterns and ensured that the corresponding MassQL queries would retrieve those scans. This was repeated for dozens of queries of different types and using different data files. From a user perspective, we use a similar approach in this manuscript when developing new, “real-world” MassQL queries. Specifically, in both the siderophore and organophosphate ester applications (highlighted in the main text) we used smaller, test datasets as “ground-truth” (i.e., known to contain the target molecules of interest) to develop and refine MassQL queries before moving to more extensive repository scale searches. In the siderophores example, we also use results from our previously-published native metabolomics workflow (see <https://doi.org/10.1038/s41557-021-00803-1>) as an orthogonal validation strategy. We highlight

this in the **“Results - Discovery of siderophores at repository scale”** section in the main text:
 “”

Combined MassQL queries for apo and bound MS2 spectra identified seven out of the eight putative siderophores identified using Ion-Identity Molecular Networking (IIMN)¹⁸ in the published analysis of the post-liquid chromatography iron addition of *E. lata* extracts. The unique compound that was found using IIMN but not by the MassQL query was missed because the ⁵⁴Fe peak intensity fell outside of the expected intensity tolerance of 25%, which is likely due to the low intensity of this peak.

“”

Moreover, in this resubmission, we highlight a query validation strategy developed by Mohanty et al. (10.1016/j.cell.2024.02.019) that used reference MS/MS spectral libraries as “ground-truth” to develop and refine MassQL queries for retrieving MS/MS spectra of bile acids. As discussed in one of the comments above, the GNPS MS/MS spectral library contains 4,533 MS/MS spectra associated with known bile acids structures. Therefore, the MS/MS spectra retrieved by each MassQL query could be compared to the MS/MS spectra expected to be retrieved. This is now described in the **“False Discovery Rate estimation and query validation strategies”** section:

“”

As demonstrated in a recent publication that used MassQL to mine LC-MS/MS raw data in the public domain and discover new, unreported bile acids³¹, one possible strategy to estimate the FDR of MassQL queries over MS/MS spectra, is to putatively identify the MS/MS spectra retrieved by MassQL using reference MS/MS spectral libraries. In their study, Mohanty et al. first used the GNPS spectral libraries (which contained 4,533 reference spectra of bile acids) to design and refine MassQL queries for bile acid spectra and estimate the queries’ selectivity. Such selectivity was measured by counting the number of retrieved bile acids and the number of retrieved non-bile acids (false positives). Thereafter, when using the refined MassQL queries to search repository data, more than 594K putative bile acids MS/MS spectra were retrieved. Of these, 270,437 MS/MS spectra were putatively identified by MS/MS library search, with 726 MS/MS matching to non-bile acids (0.27%). It is important to highlight that such FDR estimation approach is limited to the compounds that are deposited in the reference MS/MS libraries.

“”

Another potentially helpful suggestion for testing would be to organize subsets of data from MassIVE based on patterns that were reported in primary studies or patterns that were reported to be absent in the primary studies. Then the authors could search each subset for the known pattern and ensure that it is retrieved (or not retrieved) with high accuracy. Of course there may be some instances where the original papers were wrong, but I would expect these occurrences to be rare.

This is another very good suggestion. A similar strategy was indeed used by Mohanty et al. for the discovery of new, unreported bile acids published in Cell [10.1016/j.cell.2024.02.019]. As discussed in a comment above, Mohanty et al. used a subset of data from the GNPS MS/MS spectral libraries (i.e., 4,533 MS/MS spectra associated with bile acids structures) to develop and refine their MassQL queries until a suitable FDR was reached. By doing so, it was relatively

straightforward for the authors to assess the query effectiveness (i.e., counting the number of non-bile acid MS/MS spectra “wrongly” retrieved from the spectral library). We now highlight this strategy for query evaluation in the manuscript (see “**False Discovery Rate estimation and validation strategies**” section).

Reviewer #2:

Remarks to the Author:

This manuscript describes the massql package and its applications. The approach is described as a domain language to capture common query patterns in mass spectrometry data.

Although it is not described in the manuscript, the code repository suggests that the massql package uses the following Python tools: - lark for language parsing, to convert SQL style queries to JSON; - pymzml, pyteomics and matchms for data file parsing; - pandas data frames for value filtering, caching and search. The main contribution from massql appears to be the design of the language patterns.

There are many unmet demands in computational MS, driven by metabolomics and natural product research. This massql approach is a very welcome and encouraging contribution. The software package is well written, with good documentation. Search is indeed a foundational component of the field. Mass difference patterns are also fundamental to computational mass spectrometry and already included in numerous tools. Still, it is an elegant effort in massql to formalize their use.

Thanks for these supporting comments and glad the reviewer appreciates the elegant nature of the approach. We sincerely appreciate the reviewers' time to review our manuscript. The comments from both reviewers were very constructive and have significantly helped us improve our manuscript.

The authors propose the value of massql as that "The resulting MassQL language provides users the flexibility and expressiveness to query simple and complex MS patterns within their data and across public data regardless of their expertise in computational MS.

"Is massql "a universal solution to mine MS data" as claimed, or is it primarily a filtering tool for GNPS?"

This is an important question. The word "universal" was meant to emphasize the manufacturer- and platform-independent nature of the tool, meaning that data produced by any MS vendor and/or instrumental setup (e.g., LC-MS, GC-MS, direct-MS) can be queried upon conversion to mzML open format. We have revised the **Introduction** section to clarify the scope of applicability of MassQL:

“”

To address this gap, here we introduce the Mass Spectrometry Query Language (MassQL), an open-source language for flexible and mass spectrometer manufacturer-independent searching. MassQL aims to enable non-computational researchers to easily search their MS (across MS1 and MS/MS) data for patterns of interest without the need for programming skills or a dedicated computational collaborator.

“”

We agree that most of the use cases highlighted in this manuscript used MassQL within the GNPS ecosystem. Since the main goal of MassQL is to minimize the barrier of entry to MS data interrogation for non-computational scientists, we believe that the GNPS web interface offered the most accessible and user-friendly platform for the community to access the newly-introduced tool. However, the MassQL software package and workflow are open source on GitHub (<https://github.com/mwang87/MassQueryLanguage>) and can be freely downloaded and run by anyone on their own machines. Moreover, since the initial development (and submission of this manuscript), we have engaged with the wider metabolomics software community and integrated MassQL into popular metabolomics data analysis tools and infrastructure, including MZmine, pyOpenMS, MS-DIAL, UniDec, and Bruker's MetaboScape.

The following questions are relevant to this evaluation:

Q1. Does massql serve users “regardless of their expertise in computational MS”?

The massql queries are well designed and easy to use. However, the issue is what users do with the result. Based on the design, the query often returns a large number of matched patterns. As shown clearly in the examples in the manuscript, ~27K matched spectra for Figure 2 and >300K matched spectra for Figure 3. The examples are on MS2 data, while the problem with MS1 data is likely to be more challenging by multiple magnitudes. To find the few true positives from so many matches, it requires significant work and usually coding if the functions are not in GNPS.

This is a very good comment and we partially answered this in response to a similar comment from Reviewer #1. While we agree that programming skills can certainly be useful in inspecting large output files from MassQL, this cannot really be generalized as a weakness of the method for the following reasons:

- 1) The number of returned matched patterns depends on both the nature of the query and the data being queried. Very specific queries might well return a small number of matched patterns, even when run at a repository-scale. Also, we would like to stress that, although showcased in the present manuscript, repository-scale searches are not the only application of MassQL. The majority of the use cases provided in the SI and the published papers (see list below), used MassQL on individual datasets, where well designed-queries can definitely return output files that are manageable for exploration without extensive computational skills.
- 2) MassQL results are in a tabular format. Basic operations can still be performed using tools like Excel (e.g., sorting/filtering by precursor m/z or RT). Moreover, when results are inspected within the GNPS web interface, each returned match has a direct link to open the specific spectrum (or entire LC-MS file) in the GNPS dashboard for visual inspection of the matches. Also, since MassQL is supported by other data analysis software equipped with a GUI (e.g., MZmine, pyOpenMS, MS-DIAL), the entire suite of visualization tools of these software opens up for the users.
- 3) One of the key features of the NextFlow MassQL workflow is the ability to find data patterns and extract the specific MS scans in the mzML open format for downstream processing - enabling a wide range of computational downstream analysis with complementary tools. For example, MassQL can be combined with molecular networking

or spectral library matching, which can be used for result inspection/validation. For example, as illustrated in the siderophores example, MassQL can be used to retrieve spectra from a repository or a large dataset, which are then used to create a molecular network for more detailed exploration without extensive computational expertise required.

In summary, we acknowledge the reviewer's comment and agree that in some situations, especially in repository-scale searches, the number of returned matches might be hard to handle manually. However, we do believe that the MassQL ecosystem significantly lowers the barrier of entry to MS data search for non-computational scientists and reduces the need for extensive programming skills. Furthermore, the combination of MassQL with other tools (mediated by export of results in open formats) addresses concerns for downstream processing of large sets of results that can arise from less specific queries.

Q2. How to deal with false positives?

The authors are aware of this and discussed it in the Conclusion (false discovery rate etc.). But without actually addressing the issue, the scientific value of this tool is limited.

We tested massql on a LC-MS dataset of 100 human samples where a medication is known to be present in 38 of them. Massql MS1 search returned from 80 samples with positive matches of over 20 scans/spectra. Only two were returned as negatives. The false discovery rate here is unacceptable.

The authors showed in Figures 2 and 3 that significant followup work, not in massql per se, was involved in dealing with the false positives in MS2 data. But there is unlikely a workable solution for MS1 data based on massql as is.

This is a very good point, and a very similar comment was raised also by Reviewer #1. Overall we believe that the issue of FDR in MassQL cannot be addressed using a universal strategy due to the flexible nature of MassQL itself. Being a query language, the FDR depends on the nature of the data being queried, the query itself, and the expectation of what is in the result. Very specific and well-designed queries will result in lower FDR, even when used at a repository scale. We believe that tailored results validation strategies should be designed by the user, case-by-case depending on the goal of the study. This has been demonstrated by Mohanty et al. in a recent publication that used MassQL to mine public LC-MS/MS data at a repository scale and discover new, unreported bile acids (Mohanty et al. 2024, <https://doi.org/10.1016/j.cell.2024.02.019>). In their publication, the authors developed MassQL queries for retrieving bile acids of MS/MS spectra from the public MassIVE/GNPS data repository. Before the actual repository-scale search, the authors estimated the FDR of MassQL queries using the GNPS MS/MS spectral library as a reference. The GNPS MS/MS spectral library contains MS/MS spectra associated with known structural annotations (4,533 of which belong to bile acids). Therefore, the MS/MS spectra retrieved by each MassQL query could be compared to the MS/MS spectra expected to be retrieved. Mohanty et al. assessed the selectivity of each MassQL query by counting the number of non-bile acid MS/MS spectra "wrongly" retrieved from the spectral library. By doing so,

the authors could iteratively refine their queries until a suitable FDR was reached (estimated at ~0.27%) before the actual repository-scale search. The actual repository-scale search retrieved more than 270K putative bile acid MS/MS spectra from the MassIVE/GNPS repository.

We sincerely thank both reviewers for bringing up this aspect and we now added a section “**False Discovery Rate estimation and query validation strategies**” in the manuscript:

“”

False Discovery Rate estimation and query validation strategies

Overall, the key challenge when using MassQL is to define queries that are sensitive towards the target compound(s) (i.e., effectively retrieve the desired spectra), but do not retrieve too many false positive hits. Due to the flexibility and broad applicability of MassQL, a universal method for false discovery rate (FDR) estimation is difficult to establish. Rather, tailored strategies for query validation can be designed case by case by the user, depending on the research context.

As demonstrated in a recent publication that used MassQL to mine LC-MS/MS raw data in the public domain and discover new, unreported bile acids³¹, one possible strategy to estimate the FDR of MassQL queries over MS/MS spectra, is to putatively identify the MS/MS spectra retrieved by MassQL using reference MS/MS spectral libraries. In their study, Mohanty et al. first used the GNPS spectral libraries (which contained 4,533 reference spectra of bile acids) to design and refine MassQL queries for bile acid spectra and estimate the queries' selectivity. Such selectivity was measured by counting the number of retrieved bile acids and the number of retrieved non-bile acids (false positives). Thereafter, when using the refined MassQL queries to search repository data, more than 594K putative bile acids MS/MS spectra were retrieved. Of these, 270,437 MS/MS spectra were putatively identified by MS/MS library search, with 726 MS/MS matching to non-bile acids (0.27%). It is important to highlight that such FDR estimation approach is limited to the compounds that are deposited in the reference MS/MS libraries.

It should be noted that the same validation approach cannot be universally applied, for example, to the repository-scale discovery of siderophores described in the present manuscript (see **Results - Discovery of siderophores at repository scale**). Siderophore molecules do not belong to a single compound class and can exhibit very diverse chemical structures. Therefore, queries for diagnostic MS/MS fragment ions cannot be used and the search was instead performed on the MS1 level (see **Results - Discovery of siderophores at repository scale**). The adopted strategy was to first develop and refine the query on a reference dataset of *Eutypa lata* extracts known to contain iron-binding molecules¹⁷. After satisfactory results were obtained on the reference dataset (**SI Note 3.3.2**), the query was performed on a repository scale. In this specific example, we used a strategy based on a “decoy” query to get a sense for potential false discoveries (**SI Note 2**) - estimated at 21.7% for this repository scale query. Although a relatively large number of false positive hits may be expected when performing searches at a repository scale, these false positives can be mitigated by utilizing additional computational tools in a downstream analysis (e.g., molecular networking, spectral library search) to validate the query results. In these cases, MassQL acts more as a pre-filter to reduce the data to a more tractable size (from hundreds of millions of spectra to a few thousands) and “enrich” them with putative leads for further investigation and confirmatory experiments (see **Results - Discovery of siderophores at repository scale** and **Organophosphate esters in the environments**).

Overall, we encourage users to critically inspect query results and develop tailored validation strategies that are fit for their research purposes.

“””

It must be noted that the same FDR estimation approach described above cannot be used when performing searches at the MS1 level. For that, Reviewer #1 suggested a potential strategy for assessing the FDR of MS1-level searches, which uses a “decoy” query for a ^{13}C isotopic pattern that is very unlikely to occur - i.e., ^{13}C isotope peak larger than the ^{12}C isotope peak in the MS1 spectrum. We used the repository-scale search of siderophores (described in the main text) as a representative example and included the results of this strategy in **SI Note 2 - False Discovery Rate estimation of siderophores at repository scale**:

“””**SI Note 2 - False Discovery Rate estimation of siderophores at repository scale**

In the main manuscript, we discuss a possible strategy to assess FDR of MassQL searches at the MS/MS level, as demonstrated by Mohanty et al.¹ (see **False Discovery Rate estimation and query validation strategies**). However, the same FDR estimation strategy can not be used when performing searches at the MS1 level. A potential strategy for assessing the FDR of an MS1 query is to create a “decoy” query that includes isotopic patterns that are unlikely to occur for the target chemical or compound class under investigation. In the specific case of the siderophore query, we created a “decoy” query that includes an unlikely $^{12}\text{C}/^{13}\text{C}$ isotopic profile - i.e., ^{13}C isotope peak larger than the ^{12}C isotope peak in the MS1 spectrum (**SI Figure 2.1**). This is very unlikely to occur for singly charged organic compounds below 1500 Da, especially for natural products. We ran the following “true” and “decoy” MassQL queries on all public high-resolution Thermo Fisher Q Exactive data available in the GNPS/MassIVE repository and compared the results.

“True” MassQL Query:

```
QUERY scaninfo(MS1DATA) WHERE MS1MZ=X-
1.993:INTENSITYMATCH=Y*0.063:INTENSITYMATCHPERCENT=25:TOLERANCEPPM=10 AND
MS1MZ=X:INTENSITYMATCH=Y:INTENSITYMATCHREFERENCE:INTENSITYPERCENT=5 AND
MS1MZ=X+1:INTENSITYMATCH=Y*0.5:INTENSITYMATCHPERCENT=60 AND MS1MZ=X-
52.91:TOLERANCEPPM=10 AND MS2PREC=X-52.91
```

“Decoy” MassQL Query:

```
QUERY scaninfo(MS1DATA) WHERE MS1MZ=X-
1.993:INTENSITYMATCH=Y*0.063:INTENSITYMATCHPERCENT=25:TOLERANCEPPM=10 AND
MS1MZ=X:INTENSITYMATCH=Y:INTENSITYMATCHREFERENCE:INTENSITYPERCENT=5 AND
MS1MZ=X+1:INTENSITYMATCH=Y*2.0:INTENSITYMATCHPERCENT=60 AND MS1MZ=X-
52.91:TOLERANCEPPM=10 AND MS2PREC=X-52.91
```

SI Figure 2.1 - Graphical representation of the “true” (A) and “decoy” (B) MassQL queries. Green peaks represent the MS1 pattern being searched by the query. In the “decoy” query, a ^{13}C isotope peak larger than the corresponding ^{12}C is being searched. Plots were created using the MassQL sandbox (<https://massql.gnps2.org/>).

MassQL returned 749,791 and 644,108 MS1 spectra for the “true” query and the “decoy” query, respectively. This resulted in an overall estimated FDR of 85.9%. However, the m/z distribution of the returned MS1 hits by both queries are heavily skewed towards low m/z values (**SI Figure 2.2A**). Since microbial siderophores reported in literature predominantly lie in the mass range of 500-1500 Da², we restricted the count to this m/z range (**SI Figure 2.2B**). By doing so, the number of MS1 matches are 107,445 and 23,421 for the “true” and “decoy” query, respectively. This results in an estimated FDR of 21.7%.

SI Figure 2.2 - Distribution of m/z associated with the MS1 hits returned by the “true” and “decoy” MassQL queries (histogram bins of 1Da). Each returned MS1 spectrum is associated with the m/z matching the specific query pattern (i.e., X variable in the MassQL query). Distribution of m/z for the “true” and “decoy” query are displayed in blue and orange, respectively. A) Results considering the full 100-1500 m/z range (FDR=85.9%); B) Results considering the 500-1500 m/z range.

In general, we acknowledge that running MassQL queries at the MS1 level most likely leads to higher FDR (although depending on the specificity of the query). Addressing the FDR question for MS1 queries is difficult. However, our perspective is that even in the absence of an FDR

estimation/control to a low level, MassQL still provides value even for MS1 level searches. For example, in specific applications, MassQL can be used as a pre-filtering tool for very big datasets (e.g., data repositories), which reduces the data amount to a tractable size and “enrich” for useful leads, prior to further downstream analysis (e.g., molecular networking). For example, in the organophosphate ester use case in the main text, performing the same repository-scale analysis without MassQL would require building a molecular network from all public data (which is currently computationally prohibitive). By using MassQL, we greatly reduced the data size and made molecular networking possible, which helped to focus attention on families of specifically organophosphate molecules.

Regarding the example mentioned by the reviewer (i.e., LC-MS dataset of 100 human samples), it is difficult to evaluate and assess the issue given the details provided by the reviewer. False positives can be heavily influenced by the specificity of the query for the molecules of interest and this is especially so for MS1 queries. If the target molecules for the reviewer did not have unique isotopic patterns, it might be useful to utilize other MassQL language features such as other MS1 peaks that indicate expected adducts, unique MS/MS fragments for the molecules at hand, retention time and/or ion mobility bounds, minimum MS1 intensity, and MS1 mass tolerances. We sincerely thank the reviewer for trying out MassQL but without details it is difficult to speak broadly on how to optimize their specific query for their goals. However, we have updated the text to provide better characterization of FDR issues and some representative strategies for assessment going forward (see text above).

Q3. How good is the scalability of massql?

The authors did not describe if the search against GNPS repository is file by file, or using existing infrastructure, e.g. indexing data or database. They acknowledged in the Conclusion, “[...] each MassQL query searching across a billion analytes requiring hours of compute time”. This statement is likely on MS2 data. Many MS2 search tools already exist. Flash entropy MS2 search is very fast (<https://doi.org/10.21203/rs.3.rs-2693233/v1>). How does massql compare to the existing tools?

This is a good comment and also asked by reviewer #1. MassQL and MS/MS search tools (e.g., spectral library match, MASST) are fundamentally different. MS/MS search tools are constrained and limited to utilizing the full MS/MS spectrum to retrieve a match. However, MassQL provides users with the precision and flexibility to leverage their knowledge of characteristic fragmentation patterns for the compound class of interest. For example, it is known that small structural modifications can result in large changes in the overall fragmentation patterns, causing potentially relevant molecules to evade discovery by similarity-based search. This can be avoided by designing MassQL queries that specifically search for the presence (or absence) of key fragments or neutral losses in the MS/MS spectrum, regardless of the overall similarity. As an example, Selegato et al. show how MassQL could retrieve spectra of beauvericins analogs that could not be discovered by Feature-Based Molecular Networking (an approach that utilizes full MS/MS similarity) alone due to difference in fragmentation patterns. (<https://doi.org/10.3389/fmolb.2023.1238475>). In addition, information about the

precursor ion information, retention, and drift time, as well as any combination thereof can be included as additional query constraints. Thus, MassQL is complementary to existing MS2 matching tools as it gives more flexibility to the user to utilize their own knowledge of characteristic fragmentations of the compound class under investigation and encode this knowledge into a MassQL query. Given the consensus that this is a common question from both reviewers, we have addressed this aspect in more details in the **Discussion** section:

“”

While traditional MS/MS similarity/search tools are a powerful technique for matching related or similar compounds within libraries or repositories (e.g., spectral library search⁴³, MASST⁴⁴), MassQL provides a complementary set of capabilities with enhanced flexibility and precision. Specifically, traditional MS/MS search tools rely on a full MS/MS similarity measure to retrieve a match, whereas MassQL queries are a set of user-defined constraints to retrieve a spectrum. This provides the flexibility to search for more specific and complex patterns (e.g., combine MS1 and MS/MS patterns, retention and drift time constraints, see **SI Note 3.3** and **3.4**) and empowers scientists to leverage their domain knowledge of the chemical or compound class under investigation. A specific example where MassQL can complement MS/MS similarity is when small structural modifications can result in large changes in the overall fragmentation patterns. This situation can cause relevant analog molecules to evade discovery by MS/MS similarity-based search. MassQL has been shown to complement MS/MS comparison strategies by enabling the searching for conserved key fragments or neutral losses in the MS/MS spectrum without requiring a full MS/MS similarity match (for example in Selegato et al.⁴²).

“”

Regarding the scalability of MassQL itself, we acknowledge that query speed is not the main innovation in the MassQL reference implementation and there is always substantial room for optimization. The reference implementation of MassQL is primarily geared as a point of reference for semantic correctness for MassQL queries. Since the MassQL grammar and query engine are two separate components in the MassQL ecosystem, new query engines can be developed for improved search speed, while using the same MassQL formal grammar. This has already happened for the popular software packages MS-DIAL, MZmine and Bruker's MetaboScape, which implemented their own query engine. We now clarify this aspect in the new **“SI Note 1 - MassQL query speed evaluation”** :

“”

SI Note 1 - MassQL query speed evaluation

We carried out a MassQL performance evaluation using the reference implementation (version 0.0.15) on a standardized reference server in Amazon Web Services t3.large (Intel Xeon Platinum 8000 series processor (Skylake-SP or Cascade Lake)), 2 vCPU, 8 GB RAM, 20TB SSD storage - gp3 - 3000 IOPS 125MB/sec. We tested 5 different queries on a single mzML file containing 6827 spectra (with 3809 MS/MS) from Thermo Fisher Q-Exactive mass spectrometer (https://massive.ucsd.edu/ProteoSAFe/DownloadResultFile?forceDownload=true&file=f.MSV00_0086206/ccms_peak/raw/S_N1.mzML). We divided queries into two categories: simple (n=4) and complex (n=1). Simple queries searched for 1-2 peaks (or neutral losses) in MS/MS spectra. The complex query searched for a mass delta pattern (+164.9, +329.8 Da) between any 3 peaks in the MS/MS spectra. These queries and corresponding measured query times (average across 10 trials) using a pre-processed cache version of the spectra are reported in **SI Table 1**.

Simple queries were run in between 0.2 and 0.3 seconds, whereas the complex query took 148 seconds to complete. “Simple” broadly means searching for specific masses or neutral losses and not using variables (e.g., patterns between any possible m/z (s) in the spectrum), which are a complex language feature. These complex language features are expected to increase query times on the reference implementation as this reference has not been optimized specifically to accelerate these complex queries.

New optimized query engines can be developed independently and use the MassQL grammar, and potentially improve query speed both in terms of computing efficiency. For example, the MassQL language has already been implemented in various MS data analysis software (MS-DIAL, MZmine, and Bruker’s MetaboScape), which use their own query engine, while keeping the same MassQL formal grammar.

Query	Query type	Query time (s)
QUERY scaninfo(MS2DATA) WHERE MS2PROD=123.0417:TOLERANCEPPM=10	Simple	0.28 ± 0.07
QUERY scaninfo(MS2DATA) WHERE MS2PROD=123.0417:TOLERANCEPPM=10 AND MS2PROD=121.0624:TOLERANCEPPM=10	Simple	0.30 ± 0.04
QUERY MS2DATA WHERE MS2PROD=226.18 AND MS2PREC=226.1797	Simple	0.20 ± 0.03
QUERY scannum(MS2DATA) WHERE MS2NL=163	Simple	0.22 ± 0.04
QUERY scaninfo(MS2DATA) WHERE MS2PROD=X:TOLERANCEMZ=0.1:INTENSITYPERCENT=5 AND MS2PROD=X+164.9:TOLERANCEMZ=0.1:INTENSITYPERCENT=5 AND MS2PROD=X+329.8:TOLERANCEMZ=0.1:INTENSITYPERCENT=5	Complex	148.82 ± 2.35

SI Table 1 - Summary of MassQL query speeds. This table demonstrates the MassQL query speed over an evaluation LC-MS/MS file from a Q-Exactive mass spectrometer (mzspec:MSV000086206:ccms_peak/raw/S_N1.mzML). Queries are divided into “simple” and “complex” queries. Simple queries search for one or two peaks (or neutral losses) in MS/MS spectra. The complex query searches for a mass delta pattern between any three peaks in the MS/MS spectra. All run times are performed on an AWS t3.large virtual machine. Measurement of query time uses a pre-processed cached version of the spectra to avoid expensive initial processing of the mzML files.

“””

Additionally, we have added a **Methods** section describing the details of the reference implementation as a guide to both readers and developers (text reported in the answer to the next reviewer’s comment). With that being said, the reference implementation of MassQL has already

proven practically usable for discovery tasks, although not being explicitly aimed for speed. For example, for the organophosphate example in the main text, the repository-scale query used a total of **810.1** CPU-hours (~12 wall-hours on a modern desktop workstation). This does not include the engineering optimizations we implemented in the MassQL reference implementation since the original submission (discussed in more detail in the response to the next comment). We expect the updated reference implementation to be significantly faster, which is tractable on a modern commodity workstation with 32-64 CPU cores.

The MS1 search we did in Q2 took about 2 seconds per mzML file for querying a single m/z value. When they were queried for an isotope mass delta (as Supplemental example 2.1), each file took ~20 seconds. That is, to perform a single query of one isotopic mass difference on 1 million MS1 data files may take 200 days on a single CPU core. If the scalability stated in the manuscript depends on infrastructures (pre-existing database, or expensive hardware), this should be explained.

Here too, we agree with the reviewer that it is important to include more details about the existing MassQL computational ecosystem and its practicality. We have now added a **Methods** section describing the details of the reference implementation of MassQL as a guide to both readers and developers:

“””

Methods

MassQL reference implementation

- The reference implementation is a fully working version of the MassQL software ecosystem for the community to use. It also serves as a guide for future MassQL implementations that may optimize speed and/or introduce new functions in other systems. Specifically, the reference implementation includes the following pieces:
 - • MassQL formal grammar. The grammar is defined using the extended backus-naur form and builds upon common MS terminology for improved expressiveness and interpretability (see **MassQL Language Description** section). During the development, community input and feedback shaped the vocabulary and capabilities of the language.
 - • MassQL parser. The parser transforms a query into an internal data structure that can be used by any programming language. The parsing is done by using the lark Python library (<https://github.com/lark-parser/lark>) and specific Python code to transform a MassQL query to a parse tree and into the internal data structure that organizes all query conditions and qualifiers.
 - • MassQL query engine. The MassQL reference query engine is written in Python and utilizes pyteomics⁴⁵ to read open mass spectrometry data files from mzML, mzXML, and MGF formats into data frames. Such MS spectra in data frame format can optionally be saved as Apache feather files to cache data for repeated querying. The query engine itself processes the query over these data frames using the Python pandas library to perform data filtering and manipulations. Output results are data frames that can be exported as a tabular format. Optionally, the retrieved MS spectra can be exported in JSON format, MGF, and mzML⁴⁶.

- MassQL NextFlow workflow. The MassQL reference NextFlow⁴⁷ workflow is designed as an automated high-throughput tool for querying multiple files simultaneously on a computational cluster. This workflow utilizes the reference query engine and parallelizes the querying of multiple files across a multi-core processor or a batch cluster, depending on the compute resources available. All results are then merged together, including extracted MS spectra in JSON format, MGF, and mzML format, if desired.

“””

Additionally, we have carried out a performance evaluation of MassQL and added the results in the “**SI Note 1 - MassQL query speed evaluation**” (text reported in the answer to the previous comment). With that being said, we would like to highlight that, since the original submission, we have improved compute speed through engineering optimizations for the reference implementation. For instance, in the MassQL reference implementation, a significant portion of the compute time (>98% in simple queries, like the ones described here) was simply reading the mass spectrometry data rather than actually computing the queries. Moreover, we have included a NextFlow workflow that enables querying of multiple files in parallel. The compute requirements are expected to scale linearly to the number of MS data experimental files that are provided.

As discussed in the answer to the previous comment, we acknowledge that query speed is not the main innovation in the MassQL reference implementation and there is always substantial room for optimization. Nonetheless, the reference implementation of MassQL has already proven practically usable for discovery tasks and new, optimized query engines are being already developed by popular MS data analysis software packages such as MS-DIAL, MZmine and Bruker’s MetaboScape (see answer to previous comment)

We found no magic in the search functions in the source code of massql. It appears to depend on functions from pandas data frame. If that’s the case, any developer should achieve similar performance by quick scripting.

We thank the reviewer for this comment and a very similar point was raised by reviewer #1. We agree that it is possible for a computational scientist to write bespoke software to accomplish the same capabilities of a MassQL query. However, this task may not be trivial for researchers lacking a training in computer science (i.e., the vast majority of chemists and biologists). The main goal of MassQL is to lower the barrier to entry for chemists, biologists, and mass spectrometry scientists without programming skills to search their own MS data for patterns of interest. The conciseness and expressiveness of the MassQL language, which builds upon common mass spectrometry (MS) terminology, make MassQL queries easy to understand and alter for scientists with basic familiarity with MS. This empowers non-computational scientists with the ability to leverage their domain knowledge of the chemical (or compound class) under investigation, without the need for programming skills or a dedicated computational collaborator. To further increase accessibility to the MassQL ecosystem for non-computational scientists, we have created extensive accompanying infrastructure (e.g., MassQL documentation, MassQL Compendium, MassLQ sandbox for query testing, and the new addition of a MassQL Chatbot), which altogether makes writing MassQL queries much more straightforward for non-

computational users. We aimed at demonstrating this with the large number of use cases presented in the Supplementary material, where several scientists with different backgrounds (and limited computational experience) were able to develop a wide variety of MassQL queries and use them to explore their own MS data. We are pleased to see that some showcase examples have now turned into actual publications and many other manuscripts that used MassQL to make discoveries have been published over that past year (see list below). We believe this demonstrates the utility and usability of the tool for non-computational scientists. List of publications that used MassQL:

- Discovery of new bile acids at repository scales (*Cell*)
<https://doi.org/10.1016/j.cell.2024.02.019>
- Mining of lipid changes in parasitic disease (*Nature Communications*, adapted from **SI Note 4.3**)
<https://doi.org/10.1038/s41467-023-42247-w>
- Discovery of a fungal pentose-binding biotransformation enzyme (*PNAS*)
<https://doi.org/10.1073/pnas.2301007120>
- Biomonitoring of polyphenols in urine using MassQL (*Analytical Chemistry*)
<https://doi.org/10.1021/acs.analchem.3c01393>
- Link biosynthetic gene clusters and MassQL output (*PNAS Nexus*)
<https://doi.org/10.1093/pnasnexus/pgac257>
- Mining of heteromeric adducts in public datasets using MassQL (*Analytica Chimica Acta*)
<https://doi.org/10.1016/j.aca.2022.340352>
- Mining of large plant metabolomics datasets (*GigaScience* and *Nature Scientific Data*):
<https://doi.org/10.1093/gigascience/giac124>
<https://doi.org/10.1038/s41597-024-03094-6>
- MassQL-based exploration of molecular networks for natural product discovery
<https://doi.org/10.1021/acs.jnatprod.3c00750> (*Journal of Natural Products*)
<https://doi.org/10.1021/acscentsci.3c00800> (*ACS Central Science*)
- MassQL-integrated molecular networking for studying MS/MS fragmentation patterns of beauvericin analogs (*Frontiers in Molecular Biosciences*)
<https://doi.org/10.3389/fmolb.2023.1238475>
- Screening of copper adducts (*Journal of Natural Products*)
<https://doi.org/10.1021/acs.jnatprod.4c00049>
- Discovery of cupriachelin analogs (*Frontiers in Chemistry*, originally **SI Note 4.10**)
<https://doi.org/10.3389/fchem.2023.1256962>

We sincerely thank the reviewer for this comment as it gave us the possibility to clarify these points throughout the main text (see **Introduction**, **Results** and **Discussion** section) and better tailor the manuscript to our target audience (i.e., non-computational scientists).

Q4. What is the value that massql brings to MS2 search? What's different from existing MS2 tools? If a spectrum is treated as a spectrum without considering the MS level, what's different from other MS search methods? Can the value be demonstrated outside GNPS?

This is a good question, asked also by reviewer #1. MassQL and MS/MS search tools (e.g., spectral library match, MASST) are fundamentally different. MS/MS search tools are constrained and limited to utilizing the full MS/MS spectrum to retrieve a match. However, MassQL provides users with the flexibility to leverage their knowledge of characteristic fragmentation patterns for the compound class of interest. For example, it is known that small structural modifications can result in large changes in the overall fragmentation patterns, causing potentially relevant molecules to evade discovery by similarity-based search. This can be avoided by designing MassQL queries that specifically search for the presence (or absence) of key fragments or neutral losses in the MS/MS spectrum, regardless of the overall similarity. As an example, Selegato et al. shows how MassQL could retrieve spectra of beauvericins analogs that could not be discovered by Feature-Based Molecular Networking (an approach that utilizes full MS/MS similarity) alone due to difference in fragmentation patterns.(10.3389/fmolb.2023.1238475). In addition, information about the precursor ion information, retention, and drift time, as well as any combination thereof can be included as additional query constraints.

Thus, MassQL is complementary to existing MS2 matching tools as it gives more flexibility to the user to utilize their own knowledge of characteristic fragmentations of the compound class under investigation and encode this knowledge into a MassQL query. Given the consensus that this is a common question from both reviewers, we have addressed this aspect in more details in the **Discussion** section

“””

While traditional MS/MS similarity/search tools are a powerful technique for matching related or similar compounds within libraries or repositories (e.g., spectral library search⁴³, MASST⁴⁴), MassQL provides a complementary set of capabilities with enhanced flexibility and precision. Specifically, traditional MS/MS search tools rely on a full MS/MS similarity measure to retrieve a match, whereas MassQL queries are a set of user-defined constraints to retrieve a spectrum. This provides the flexibility to search for more specific and complex patterns (e.g., combine MS1 and MS/MS patterns, retention and drift time constraints, see **SI Note 3.3** and **3.4**) and empowers scientists to leverage their domain knowledge of the chemical or compound class under investigation. A specific example where MassQL can complement MS/MS similarity is when small structural modifications can result in large changes in the overall fragmentation patterns. This situation can cause relevant analog molecules to evade discovery by MS/MS similarity-based search. MassQL has been shown to complement MS/MS comparison strategies by enabling the searching for conserved key fragments or neutral losses in the MS/MS spectrum without requiring a full MS/MS similarity match (for example in Selegato et al.⁴²).

“””

Regarding the value of MassQL outside the GNPS environment, we believe that the GNPS ecosystem offered the most accessible and user-friendly platform for the community to access the newly-introduced tool (at the time of the original submission). As discussed in one of the comments above, since the initial development and submission of this manuscript, we have engaged with the wider metabolomics software community and integrated MassQL into popular metabolomics data analysis tools (MZmine, pyOpenMS, MS-DIAL, UniDec, Bruker's MetaboScape) and infrastructure (Metabolomics Workbench).

Q5. What is the value that massql brings to MS1 search?

As discussed in Q2, many false positives are expected by this approach. The MS2 data usually contain a higher ratio of useful information than MS1, because human hypotheses were often driving their collection in the first place. In untargeted MS1 data from complex samples, the scan-level search as in massql now bears no confidence. It will be too erroneous and too slow, so that we see no value in using massql to mine MS1 data.

We thank the reviewer for this question and a very similar concern was also raised in by reviewer #1. While it is potentially true that the amount of information contained in MS2 spectra may be larger compared to MS1 data, we believe that the nature of this information can be complementary.

SI Note 3.1 - MassQL Query Using MS1 Spectral Information shows specific applications where MS1 data were mined using MassQL. In particular, **SI Note 3.1.1 - Tracking The Biosynthesis of Isotopically Labeled Phenylpropanoids** shows an example of MS1 data mining in isotope labeling experiments and **SI Note 3.1.2 - Chlorinated Compounds in Lichen Thalli Extracts** show examples of MS1 data mining for finding mono-, di-, and tri-chlorinated compounds from complex natural extracts. Similar queries leveraging characteristic isotopic signatures (e.g., bromine, sulfur) can also be designed and, possibly made more specific by including retention and drift time filters.

In addition to this, MS2 searches can be combined with patterns in the MS1 data (e.g., presence of a characteristic isotopic pattern for the precursor m/z) to make the MassQL queries more specific, compared to simpler MS2 searches. Examples of these types of queries are now reported in **SI Note 3.3 - MassQL Query Using MS1 and MS/MS Spectral Information**.

Finally, as discussed in the answer to Q2, MassQL can be used on the MS1 level as a pre-filtering tool for very big datasets (e.g., data repositories) in order to reduce the data amount to a tractable size and “enrich” for useful leads, prior to further downstream analysis (e.g., molecular networking).

Q6. What are the alternatives to massql?

Many functions in massql can be performed by custom scripting. In our group, when we need to search for a neutral loss in the data, we write a quick script and get it done. It is understood that massql enables many more people to do queries, given that the queries are still a long way from discoveries and their values are still debatable. In this regard, massql occupies a unique niche.

The reviewer is correct and we are glad they appreciate the uniqueness of MassQL. As discussed in the answer to one of the comments above, we agree with the reviewer that it is possible for a computational scientist to write bespoke software to accomplish the same capabilities of a MassQL query. However, this task may not be trivial for researchers lacking a training in computer

science (i.e., the vast majority of chemists and biologists). The main goal of MassQL is to lower the barrier to entry for chemists, biologists, and mass spectrometry scientists without programming skills to search their own MS data for patterns of interest. The conciseness and expressiveness of the MassQL language, which builds upon common mass spectrometry (MS) terminology, make MassQL queries easy to understand and alter for scientists with basic familiarity with MS. This empowers non-computational scientists with the ability to leverage their domain knowledge of the chemical (or compound class) under investigation, without the need for programming skills or a dedicated computational collaborator.

We are very happy with the interactions with both reviewers as it gave us the possibility to clarify this point throughout the main text (see **Introduction**, **Results** and **Discussion** section) and better tailor the manuscript to our target audience (i.e., non-computational scientists).

There are many tools for isotope analysis. Massql offers no perceivable advantage over existing tools.

We thank the reviewer for this comment. We agree that many tools for isotope analysis exist. An important example is the *Chelomics* tool (from the Baars group) for finding iron-chelating compounds. While we believe *Chelomics* is a great and fit-for-purpose tool, we also believe that it would be very difficult for non-computational scientists to change the behavior of the software (e.g., change the MS pattern being searched). As shown in **SI Note 2 - False Discovery Rate estimation of siderophores at repository scale**, the design of a “decoy” query to assess the FDR of the repository-scale search of siderophores (suggested by reviewer #1) required only a simple change in the MassQL query. In addition, *Chelomics* requires MATLAB, a proprietary software, in order to operate. This limits the implementation of the tool into open source software and platforms.

With that being said, while one of the examples highlighted in the main manuscript (i.e., **Discovery of siderophores at repository scale**) uses MassQL to search for MS1 patterns, this is just one of the many possible applications of MassQL. In fact, isotope patterns in the MS1 can be used in conjunction with other MS data pattern filters (e.g., MS1 + MS2 patterns, inclusion of retention time and/or ion mobility constraints) to make queries more specific towards the chemical or compound class of interest. Performing this combination using multiple specialized software tools (e.g., the *Chelomics* tool mentioned above) would require the user to learn how to use them individually, and will most likely require custom scripting to combine the different outputs. A key advantage of MassQL is the possibility and flexibility for non-programmers to easily combine all these MS data patterns into a concise and expressive query. Finally, in terms of visualization and exploration of the output results, MassQL can be efficiently combined with further downstream analysis tools (e.g., spectral library search, molecular networking) with a few clicks within the GNPS environment. Outside the GNPS environment, MassQL is already supported by popular MS data analysis packages equipped with a GUI (e.g., MZmine, MS-DIAL), where the entire suite of visualization tools of these software opens up for the users.

Will massql be the tool of choice for mining mass spectrometry data (independent from GNPS)?

As the reviewer mentioned in Q6, we believe that MassQL occupies a unique niche in the computational metabolomics landscape. The large usability of the tool now empowers non-computational scientists to leverage their domain-specific knowledge and flexibly search for MS data patterns of interest without the need for programming skills or a dedicated computational collaborator. This is demonstrated by the large diversity of research fields and contexts MassQL has already been applied to (even prior to its publication). Besides the large number of use cases reported in the Supplementary material of this manuscript (some of which already turned into actual publications), many other manuscripts that used MassQL to make discoveries have been published over that past year:

- Discovery of new bile acids at repository scales (*Cell*)
<https://doi.org/10.1016/j.cell.2024.02.019>
- Mining of lipid changes in parasitic disease (*Nature Communications*, adapted from **SI Note 4.3**)
<https://doi.org/10.1038/s41467-023-42247-w>
- Discovery of a fungal pentose-binding biotransformation enzyme (*PNAS*)
<https://doi.org/10.1073/pnas.2301007120>
- Biomonitoring of polyphenols in urine using MassQL (*Analytical Chemistry*)
<https://doi.org/10.1021/acs.analchem.3c01393>
- Link biosynthetic gene clusters and MassQL output (*PNAS Nexus*)
<https://doi.org/10.1093/pnasnexus/pgac257>
- Mining of heteromeric adducts in public datasets using MassQL (*Analytica Chimica Acta*)
<https://doi.org/10.1016/j.aca.2022.340352>
- Mining of large plant metabolomics datasets (*GigaScience* and *Nature Scientific Data*):
<https://doi.org/10.1093/gigascience/giac124>
<https://doi.org/10.1038/s41597-024-03094-6>
- MassQL-based exploration of molecular networks for natural product discovery
<https://doi.org/10.1021/acs.jnatprod.3c00750> (*Journal of Natural Products*)
<https://doi.org/10.1021/acscentsci.3c00800> (*ACS Central Science*)
- MassQL-integrated molecular networking for studying MS/MS fragmentation patterns of beauvericin analogs (*Frontiers in Molecular Biosciences*)
<https://doi.org/10.3389/fmolb.2023.1238475>
- Screening of copper adducts (*Journal of Natural Products*)
<https://doi.org/10.1021/acs.jnatprod.4c00049>
- Discovery of cupriachelin analogs (*Frontiers in Chemistry*, originally **SI Note 4.10**)
<https://doi.org/10.3389/fchem.2023.1256962>

Regarding the value of MassQL outside GNPS, as discussed in Q4, the GNPS ecosystem initially provided the most accessible and user-friendly platform for the community to access the newly-introduced tool (at the time of the original submission). Since then, we have engaged with the broader metabolomics software community and integrated MassQL into several popular metabolomics data analysis tools (MZmine, pyOpenMS, MS-DIAL, UniDec, Bruker's MetaboScope) and infrastructure (Metabolomics Workbench). We believe this wide adoption by the metabolomics software community is significant for a yet-unpublished tool, and we envision

the MassQL computational ecosystem will continue to grow in adoption and capability thanks to its architecture. We have also clarified this aspect in **Results** section of the manuscript:

“””

While MassQL was originally implemented within the GNPS environment^{6,9}, this was limiting for the accessibility to a broader audience. To further enhance usability, we have engaged with the wider metabolomics software community and, through these efforts, the MassQL language has been adopted and natively supported in a variety of MS data analysis software and infrastructure (**Fig. 1c**), both open-source (MZmine¹⁰, pyOpenMS¹¹, MS-DIAL¹², UniDec¹³, Metabolomics Workbench⁷) and commercial (Bruker’s MetaboScape). It must be noted that these software tools use the same MassQL language grammar but implement their own MassQL query engine backend. This provides the possibility to develop optimized query engines for improved query performance, while maintaining query semantic consistency across tools. Finally, to facilitate the integration of MassQL into other platforms and pipelines, MassQL is available as Python and R¹⁴ libraries and as a web application programming interface (API).

“””

Because the design of massql caters to SQL like queries, the benefits to developers and data scientists are not clear other than data exploration. For MS2 search, the examples like Flash entropy search are more compelling. For MS1 data search, performant examples at database scales are available. Given the lack of published examples of scientific values, we believe that mining small molecular mass spectrometry data requires significant computational expertise, more sophisticated data models than raw spectra and usually a lengthy workflow. If people need a reusable software package in such workflows, massql is unlikely to be the best choice, because search at the scan level is almost useless for MS1 and there are many reusable tools in MS2 search. Finally, we have not seen the evidence of computational performance in massql.

We thank the reviewer for these questions and perspectives as this speaks to a lack of clarity about the MassQL capabilities and goals in the original submission of the manuscript. We have now clarified in the main text that the target audience for MassQL is primarily non-computational scientists that want to flexibly search large and small MS datasets for patterns of interest. In particular, we have revised the text in both the **Introduction** section:

“””

Introduction

[...]

While bespoke one-off scripts provide the necessary flexibility to search for specific MS data patterns³, most non-computational researchers and labs lack the computational skills to develop or customize them^{4,5}. This skill gap limits biologists and chemists from effectively searching across MS datasets, potentially leaving many biologically-important molecules hidden and undiscovered in the data. To address this gap, here we introduce the Mass Spectrometry Query Language (MassQL), an open-source language for flexible and mass spectrometer manufacturer-independent searching. MassQL aims to enable non-computational researchers to easily search their MS (across MS1 and MS/MS) data for patterns of interest without the need for programming skills or a dedicated computational collaborator. In this manuscript, we describe the MassQL

language, showcase its accompanying computational ecosystem to increase accessibility, and highlight two application examples that demonstrate how MassQL can be used on entire public repositories (such as GNPS/MassIVE⁶, Metabolomics Workbench⁷, and MetaboLights⁸).

“”

and the **Discussion** section:

“”

Discussion

Here we introduced MassQL, a platform- and manufacturer-independent query language to search for MS data patterns within and across MS datasets. The main goal of MassQL is to provide non-computational scientists with the flexibility to encode complex MS patterns into concise and expressive queries, reducing the need to write bespoke programming scripts.

“”

As discussed in Q4, while we agree that tools such as Flash entropy, MASST+ and FastMASST are important contributions, they are fundamentally different from MassQL. MS/MS search tools (such as Flash entropy and MASST) are constrained and limited to utilizing the full MS/MS spectrum to retrieve a match. In contrast, MassQL queries can be seen as a set of user-defined conditions that need to be satisfied for a spectrum to be retrieved. Thus, MassQL is complementary to existing MS² matching tools as it provides more flexibility to the user to utilize their own knowledge of characteristic fragmentations of the compound class under investigation. Given the consensus that this is a common question from both reviewers, we have addressed this aspect in more details in the **Discussion** section (see also answer to Q4):

“”

While traditional MS/MS similarity/search tools are a powerful technique for matching related or similar compounds within libraries or repositories (e.g., spectral library search⁴³, MASST⁴⁴), MassQL provides a complementary set of capabilities with enhanced flexibility and precision. Specifically, traditional MS/MS search tools rely on a full MS/MS similarity measure to retrieve a match, whereas MassQL queries are a set of user-defined constraints to retrieve a spectrum. This provides the flexibility to search for more specific and complex patterns (e.g., combine MS¹ and MS/MS patterns, retention and drift time constraints, see **SI Note 3.3 and 3.4**) and empowers scientists to leverage their domain knowledge of the chemical or compound class under investigation. A specific example where MassQL can complement MS/MS similarity is when small structural modifications can result in large changes in the overall fragmentation patterns. This situation can cause relevant analog molecules to evade discovery by MS/MS similarity-based search. MassQL has been shown to complement MS/MS comparison strategies by enabling the searching for conserved key fragments or neutral losses in the MS/MS spectrum without requiring a full MS/MS similarity match (for example in Selegato et al.⁴²).

“”

Concerning the lack of published examples of scientific values, we agree that there were no publications using MassQL at the time of initial manuscript submission as the tool was just being introduced to the community. However, we are now pleased to see that, besides the large number of use cases reported in the Supplementary material of this manuscript (some of which already turned into actual publications), many other manuscripts that used MassQL to make discoveries

have been published over that past year (see list below) We believe this demonstrates the utility and usability of the tool for non-computational scientists. List of publications that used MassQL:

- Discovery of new bile acids at repository scales (*Cell*)
<https://doi.org/10.1016/j.cell.2024.02.019>
- Mining of lipid changes in parasitic disease (*Nature Communications*, adapted from **SI Note 4.3**)
<https://doi.org/10.1038/s41467-023-42247-w>
- Discovery of a fungal pentose-binding biotransformation enzyme (*PNAS*)
<https://doi.org/10.1073/pnas.2301007120>
- Biomonitoring of polyphenols in urine using MassQL (*Analytical Chemistry*)
<https://doi.org/10.1021/acs.analchem.3c01393>
- Link biosynthetic gene clusters and MassQL output (*PNAS Nexus*)
<https://doi.org/10.1093/pnasnexus/pgac257>
- Mining of heteromeric adducts in public datasets using MassQL (*Analytica Chimica Acta*)
<https://doi.org/10.1016/j.aca.2022.340352>
- Mining of large plant metabolomics datasets (*GigaScience* and *Nature Scientific Data*):
<https://doi.org/10.1093/gigascience/giac124>
<https://doi.org/10.1038/s41597-024-03094-6>
- MassQL-based exploration of molecular networks for natural product discovery
<https://doi.org/10.1021/acs.jnatprod.3c00750> (*Journal of Natural Products*)
<https://doi.org/10.1021/acscentsci.3c00800> (*ACS Central Science*)
- MassQL-integrated molecular networking for studying MS/MS fragmentation patterns of beauvericin analogs (*Frontiers in Molecular Biosciences*)
<https://doi.org/10.3389/fmolb.2023.1238475>
- Screening of copper adducts (*Journal of Natural Products*)
<https://doi.org/10.1021/acs.jnatprod.4c00049>
- Discovery of cupriachelin analogs (*Frontiers in Chemistry*, originally **SI Note 4.10**)
<https://doi.org/10.3389/fchem.2023.1256962>

Concerning the reviewer's perspective on the utility of MS1 searches, we argue that MS1 queries are indeed useful. **SI Note 3.1 - MassQL Query Using MS1 Spectral Information** shows specific applications where MS1 data were mined using MassQL. In particular, **SI Note 3.1.1 - Tracking The Biosynthesis of Isotopically Labeled Phenylpropanoids** shows an example of MS1 data mining in isotope labeling experiments and **SI Note 3.1.2 - Chlorinated Compounds in Lichen Thalli Extracts** show examples of MS1 data mining for finding mono-, di-, and tri-chlorinated compounds from complex natural extracts. Similar queries leveraging characteristic isotopic signatures (e.g., bromine, sulfur) can also be designed and, possibly made more specific by including retention and drift time filters. Moreover, as discussed in Q5, MS2 searches can be combined with patterns in the MS1 data (e.g., presence of a characteristic isotopic pattern for the precursor *m/z*) to make the MassQL queries more specific, compared to simpler MS2 searches. Examples of these types of queries are now reported in **SI Note 3.3 - MassQL Query Using MS1 and MS/MS Spectral Information**. Finally, as discussed in the answer to Q2, MassQL can be used on the MS1 level as a pre-filtering tool for very big datasets (e.g., data repositories) in order

to reduce the data amount to a tractable size and “enrich” for useful leads, prior to further downstream analysis (e.g., molecular networking).

Finally, we thank the reviewer for their comments on performance evaluation. As mentioned in Q3, we have performed a performance evaluation of the reference implementation of MassQL and reported the results in **SI Note 1 - MassQL query speed evaluation** (see answer to Q3).

Q7. What is the value of massql independent from GNPS?

The examples from the authors are mostly about prefiltering for GNPS applications. As wonderful as GNPS is, it is still a small subset of computational MS. To claim it to be “a universal solution to mine MS data”, we need to see compelling examples outside GNPS.

We thank the reviewer for their question. As mentioned above, the word “universal” was meant to emphasize the manufacturer- and platform-independent nature of the tool, meaning that data produced by any MS vendor and/or instrumental setup (e.g., LC-MS, GC-MS, direct-MS) can be queried upon conversion to mzML open format. We have revised the text in the **Introduction** section accordingly to clarify the scope of applicability of MassQL:

“”

To address this gap, here we introduce the Mass Spectrometry Query Language (MassQL), an open-source language for flexible and mass spectrometer manufacturer-independent searching. MassQL aims to enable non-computational researchers to easily search their MS (across MS1 and MS/MS) data for patterns of interest without the need for programming skills or a dedicated computational collaborator.

“”

We agree that most of the use cases presented in this manuscript used MassQL within the GNPS ecosystem. Since the main goal of MassQL was to minimize the barrier of entry to MS data interrogation for non-computational scientists, we believe that the GNPS web interface offered the most accessible and user-friendly platform for the community to access the newly-introduced tool. However, the MassQL software package and workflow are open source on GitHub (<https://github.com/mwang87/MassQueryLanguage>) and can be freely downloaded and run by anyone on their own machines. Moreover, since the initial development (and submission of this manuscript), we have engaged with the wider metabolomics software community and integrated MassQL into popular metabolomics data analysis tools and infrastructure, including MZmine, pyOpenMS, MS-DIAL, UniDec, and Bruker’s MetaboScape, Metabolomics Workbench. Notably, at the last American Society of Mass Spectrometry (June 2024), two posters highlighted MassQL as being utilized within the commercial software Bruker’s Metaboscape:

- TP 381 - Analysis of novel *Caenorhabditis elegans* specific phosphorylated glycosphingolipids using LC TIMS MS/MS and MassQL
- TP 457 - “Mass query language and collisional cross section prediction to support chemical derivatization research using mass spectrometry“

We believe this wide adoption by the metabolomics software community is significant for a yet-unpublished tool, and we envision the MassQL computational ecosystem will continue to grow in

adoption and capability thanks to its architecture. Finally, we expect a delay in scientific publications utilizing MassQL outside of GNPS because its integration in the above-mentioned software tools happened later.

We thank the reviewer for bringing up the usability of MassQL outside of the GNPS environment and we have now clarified this aspect in **Results** section of the manuscript:

“””

While MassQL was originally implemented within the GNPS environment^{6,9}, this was limiting for the accessibility to a broader audience. To further enhance usability, we have engaged with the wider metabolomics software community and, through these efforts, the MassQL language has been adopted and natively supported in a variety of MS data analysis software and infrastructure (**Fig. 1c**), both open-source (MZmine¹⁰, pyOpenMS¹¹, MS-DIAL¹², UniDec¹³, Metabolomics Workbench⁷) and commercial (Bruker's MetaboScape). It must be noted that these software tools use the same MassQL language grammar but implement their own MassQL query engine backend. This provides the possibility to develop optimized query engines for improved query performance, while maintaining query semantic consistency across tools. Finally, to facilitate the integration of MassQL into other platforms and pipelines, MassQL is available as Python and R¹⁴ libraries and as a web application programming interface (API).

“””

Q8. What is the specific value of massql in data mining?

The massql package is a nice tool for searching mass spec data. But much more work is required to get discoveries out of data mining. In all the application examples, the heavy lifting was done by other GNPS tools and the scientists.

We thank the reviewer for the comment and we agree that in the two applications highlighted in the main text of this manuscript, there is significant post-MassQL computational analysis that utilizes other GNPS tools. However this is not the only possible scenario and the combination of MassQL with downstream tools depends on the goal of the analysis and research context (primarily the scale of the data being searched). Many other application examples in the Supplementary material, and now also some recent publications (listed below), used MassQL as a standalone tool for MS data analysis. List of publications using MassQL without GNPS tools:

- Biomonitoring of polyphenols in urine (*Analytical Chemistry*)
<https://doi.org/10.1021/acs.analchem.3c01393>
- Mining of heteromeric adducts in public datasets (*Analytica Chimica Acta*)
<https://doi.org/10.1016/j.aca.2022.340352>

In other situations MassQL provides essential value when complemented with other GNPS tools. First, MassQL, highlighted in the main text examples, makes it possible to analyze very large datasets for certain classes of compounds, where MassQL is used as a pre-filter (see for example recent publication in *Cell*, <https://doi.org/10.1016/j.cell.2024.02.019>). Second, MassQL can be used as a post-processing tool to enhance the utility of existing GNPS tools, such as molecular networking. This is showcased by the publications listed below where, for example, scientists

have used MassQL to assist the investigation of molecular networking results, by prioritizing specific families of MS/MS spectra of interest:

- Discovery of a fungal pentose-binding biotransformation enzyme (*PNAS*)
<https://doi.org/10.1073/pnas.2301007120>
- Natural product drugs discovery (*ACS Central Science*)
<https://doi.org/10.1021/acscentsci.3c00800>
- Cyclic Peptides discovery (*Journal of Natural Products*)
<https://doi.org/10.1021/acs.jnatprod.3c00750>
- Depsipeptides discovery (*Frontiers in Molecular Biosciences*)
<https://doi.org/10.3389/fmolb.2023.1238475>

Finally, we believe that the fact that the “heavy lifting is done by the scientists” is indeed the main goal of MassQL, which now enables non-computational researchers to perform these searches themselves without the need for programming skills or a dedicated computational collaborator.

Overall, the authors demonstrated nice applications of massql to the GNPS system. However, this design leads to a large number of false positives, which still require significant effort and computational expertise to deal with. There is no evidence on the real scalability of the software. The claim of “a universal solution to mine MS data” is not substantiated by the data.

The targeted users of massql are unclear. The proposition is that massql removes the barrier for many people to mine mass spec data without coding knowledge. In reality, it creates a situation that is hard to get out without coding. Then the design compromises its reuse by developers.

We sincerely thank the reviewer for their time to review our manuscript and their questions and perspectives. These gave us the possibility to clarify the technical merits of MassQL and better tailor the manuscript to our target audience (i.e., non computational scientists). We have addressed the concerns about false positives, scalability and usability of MassQL outside of the GNPS ecosystem in the point-by-point answers above.

Reviewer #1:

Remarks to the Author:

The authors have responded to my comments in an impressively thorough fashion. The manuscript is much improved and all of my questions have been addressed. Since the first submission, MassQL has picked up a lot of traction in the community and there are a number of recent publications that use it. This is the best testimony for the value of the contribution, and those new papers themselves answer most of the questions I raised. It would be really nice to officially have MassQL out -- I recommend publication of the revised manuscript as is.

We thank the reviewer for their positive comments.

Reviewer #2:

Remarks to the Author:

The revised manuscript came with a lengthy rebuttal, which addresses few of the reviewers' concerns but confirms the main limitations of innovation in this work.

We thank the reviewer for their comments

1. Both reviewers stated earlier that the search functions MassQL performs were trivial, easy to do using existing tools or scripting.

The authors acknowledge this and emphasize that "the main goal of MassQL is to lower the barrier to entry for biologists and chemists without computational skills to search MS data for patterns of interest".

We agree that the previous suggestions from both reviewers and the subsequent revisions better contextualize the contributions of this work.

2. Both reviewers stressed the critical role of computational performance.

The authors acknowledge that "query speed is not the main innovation in the MassQL reference implementation". That is after they "improved the compute speed of the reference implementation of MassQL through engineering optimizations". This, with other relevant texts, means there's no innovation in query algorithms.

We agree with the reviewer and addressed this in the last round of revisions. As now mentioned in the manuscript, we recognize areas for improvement within MassQL, such as enhancing the query performance.

The table in SI Note 1 is misleading because it is based on cached data and does not represent a real-world scenario. Neither does it include comparisons to other query approaches.

We have improved the clarity of SI Note 1 and SI Table 1 - as the usage of the word cache was likely confusing. We have removed cache and clarified that we are using a pre-processed version of the spectra to potentially avoid confusion that results would be cached rather than simply the raw data.

The "scalability" claim on using NextFlow is misleading because it's throwing more hardware at the problem, not based on software performance.

We have ensured that scalability claims with regards to NextFlow are absent in the manuscript.

3. Both reviewers raised the concerns that the high number of false positives from MassQL poses a significant barrier to its practical use.

The authors have added a new section on this topic, explaining that a generic solution is not available. The example of the Mohanty paper, which is an excellent work from this group of authors, is misleading because bile acids have distinct chemical structures that are hard to confuse with other molecules.

We agree that bile acids have a distinct chemical structure in order to make it possible to search for bile acids. The examples highlighted in the SI also provide support that there exist many other classes of compounds that also have distinct chemical structures (resulting in MS1 or MS/MS patterns that could be searched for). We agree that this may not be true for all classes of molecules, and is the reason we point out that there is not a generic solution.

To help guide the audience with published examples beyond the SI applications, we have included over half a dozen published manuscripts that utilize MassQL on several different molecular classes (polyphenols, lipopeptides, plant metabolites, etc.).

The answer to the "thought experiment" proposed by Reviewer #1 is reported in SI Note 2, "MassQL returned 749,791 and 644,108 MS1 spectra for the "true" query and the "decoy" query, respectively. This resulted in an overall estimated FDR of 85.9%." The authors cut it down to 21.7% by expert knowledge that is not in MassQL per se.

This is indeed one of the goals of MassQL - MassQL can enable non-computational researchers to search their MS data and leverage their expert knowledge of the chemical or compound class under investigation - in this case adding an additional constraint around the precursor m/z.

This reviewer used in the previous evaluation an example of falsely identified drugs in a human dataset by MassQL.

While we sincerely thank the reviewer for trying out MassQL, it is difficult to evaluate and assess the issue given the lack of details provided. False positives can be heavily influenced by the specificity of the query for the molecules of interest and this is especially so for MS1 queries. If the target molecules for the reviewer did not have unique isotopic patterns, it might be useful to utilize other MassQL language features such as other MS1 peaks that indicate expected adducts, unique MS/MS fragments for the molecules at hand, retention time and/or ion mobility bounds, minimum MS1 intensity, and MS1 mass tolerances.

The problem here is that MassQL by design has no noise modeling. In MS1 data, much noise is often filtered by requiring consecutive scans in the chromatography. MassQL is a brute force approach, ignoring the nature of data. Personally, I believe it is good to acknowledge what a tool does not do. Science is better served by defining the gaps than overstating the results.

We agree that in MS1 data, approaches that utilize consecutive scans across the chromatography may aid in the reduction of noise and have been impactful for data analysis within metabolomics. We have revised the discussion section to acknowledge these limitations and agree with the reviewer that it is best to point out what MassQL does not do as well.

“””

We do acknowledge limitations with MassQL as available today – specifically MassQL has limited capabilities to leverage more than a handful of MS spectra, e.g. consecutive MS spectra arising from the elution of chromatographic peaks which can be grouped as a chromatographic feature.

“””

4. My opening question in the initial view was: Is massql “a universal solution to mine MS data” as claimed, or is it primarily a filtering tool for GNPS?

The authors have now retracted the former. The latter is clear now. Actually, it was how Dr. Dorrestein presented MassQL at the Metabolomics Society meeting this year, a filtering tool.

We agree that MassQL can be used both as a pre-filtering and standalone tool, depending on the nature of the query and the data being queried and used. For example, when querying very big datasets (e.g., data repositories), MassQL can be effectively used as a pre-filter to reduce the data amount to a tractable size prior to further downstream analysis (highlighted in applications in this manuscript). When working on non-repository datasets, MassQL can be used as a standalone tool for searching for spectra of interest. In the Supplementary material and now also some recent publications (listed below) MassQL was used as a standalone tool:

- Biomonitoring of polyphenols in urine (Analytical Chemistry)
<https://doi.org/10.1021/acs.analchem.3c01393>
- Mining of heteromeric adducts in public datasets (Analytica Chimica Acta)
<https://doi.org/10.1016/j.aca.2022.340352>

Further, MassQL can be also used as a post-processing tool to enhance the utility of existing tools, such as molecular networking. This is showcased by the publications listed below where, for example, scientists have used MassQL to assist the investigation of molecular networking results, by prioritizing specific families and MS/MS spectra of interest:

- Discovery of a fungal pentose-binding biotransformation enzyme (*PNAS*)
<https://doi.org/10.1073/pnas.2301007120>
- Natural product drugs discovery (*ACS Central Science*)
<https://doi.org/10.1021/acscentsci.3c00800>
- Cyclic Peptides discovery (*Journal of Natural Products*)
<https://doi.org/10.1021/acs.jnatprod.3c00750>

- Depsipeptides discovery (*Frontiers in Molecular Biosciences*)
<https://doi.org/10.3389/fmolb.2023.1238475>

The authors cited two papers in the rebuttal to show the value of "using MassQL without GNPS tools". However, they are also coauthors on these two papers, suggesting that these were not independent use of MassQL by the community.

While we have been very excited about the adoption of MassQL within the GNPS ecosystem, as evidenced by the dozens of publications prior to publication. Beyond the two of the publications listed (that included the corresponding author of this manuscript as co-author), there were several recent instances of MassQL being used outside of GNPS and where the corresponding author was not a co-author of the manuscript. At the last American Society of Mass Spectrometry (June 2024), two posters highlighted MassQL as being utilized within the commercial software Bruker's Metaboscape where the corresponding author was not a co-author:

- TP 381 - Analysis of novel *Caenorhabditis elegans* specific phosphorylated glycosphingolipids using LC TIMS MS/MS and MassQL
- TP 457 - "Mass query language and collisional cross section prediction to support chemical derivatization research using mass spectrometry"

Additionally, in the ASMS conference in June of 2023, two posters highlighted MassQL where the corresponding author was not a co-author:

- ThP 521 - "Flexible, Vendor-agnostic Assessment of Liquid Chromatography - Mass Spectrometry System Performance using MassQL"
- ThP 350 - "SpectraSpectre: An implementation of MassQL for rapid querying of MS data"

We acknowledge that these are peer reviewed publications. We hypothesize that MassQL in GNPS is likely the most user-friendly place to use MassQL - thus making it the first place many researchers would have tried it out. We expect a delay in scientific publications utilizing MassQL outside of GNPS because its integration in the above-mentioned software tools happened after the initial implementation in GNPS - but the posters in the last two ASMS conferences demonstrate evidence of utilization of MassQL outside of GNPS where the corresponding author is not a co-author.

The Mohanty paper by this group of authors is a great example of how reanalysis of repository data led to new discoveries. Could the Mohanty paper be accomplished effectively without MassQL? Absolutely yes.

We thank the reviewer finding that repository scale queries enabled by MassQL was a great example of how new discoveries are possible with the computational approaches introduced in this manuscript.

Overall, the authors responded to the reviewers' concerns with mostly negative results. This manuscript, as currently written, is like a protocol paper not a method paper. A method paper needs to prove its novelty and performance in the right scientific context. Most examples and statements here are bundled to the popularity of GNPS. While MassQL is a valuable enhancement to GNPS, its significance shall be evaluated on its standalone scientific contributions, which appear to be limited.

We thank the reviewer for their perspective and agree that MassQL is a valuable contribution. We also agree that as presented the scientific discoveries made with MassQL inside of GNPS are significantly greater than outside of GNPS. We hypothesize this is due to the fact that MassQL was first introduced within GNPS and that is likely the most user-friendly place to use the tool. However, the wide adoption of MassQL in open source software, vendor analysis software, and even metabolomics repositories sets a solid foundation for MassQL to be used broadly across the metabolomics field.